



# Observing the Evolution of Summer Melt on Multiyear Sea Ice with ICESat-2 and Sentinel-2

Ellen M. Buckley[1,2], Sinéad L. Farrell[2,3], Ute C. Herzfeld[4], Melinda Webster[5,6], Thomas Trantow[4], Oliwia N. Baney[3], Kyle A. Duncan[7], Huilin Han[4], and Matt Lawson[4]

[1]Center for Fluid Mechanics, Brown University, Providence, RI, USA
[2]Department of Atmospheric and Oceanic Sciences, University of Maryland, College Park, MD, USA
[3]Department of Geographical Sciences, University of Maryland, College Park, MD, USA
[4]Department of Electrical, Energy and Computer Engineering, University of Colorado, Boulder, CO, USA
[5]Geophysical Institute, University of Alaska Fairbanks, Fairbanks, AK, USA
[6]Polar Science Center, University of Washington, Seattle, WA, USA
[7]Earth System Science Interdisciplinary Center, University of Maryland, College Park, MD, USA

**Correspondence:** Ellen M. Buckley (buckley@umd.edu)

**Abstract.** We investigate sea ice conditions during the 2020 melt season, when warm air temperature anomalies in Spring led to early melt onset, an extended melt season and the second-lowest September minimum Arctic ice extent observed. We focus on the region of the most persistent ice cover and examine melt pond depth retrieved from ICESat-2 using two distinct algorithms in concert with a time series of melt pond fraction and ice concentration derived from Sentinel-2 imagery to obtain

insights about the melting ice surface in three dimensions. We find melt pond fraction derived from Sentinel-2 in the study region increased rapidly in June, with the mean melt pond fraction peaking at 16% +/- 6% on 24 June 2020, followed by a slow decrease to 8% +/- 6% by 3 July, and remained below 10% for the remainder of the season through 15 September. Sea ice concentration was consistently high (>95%) at the beginning of the melt season until 4 July, and as floes disintegrated, decreased to a minimum of 70% on July 30, then became more variable ranging from 75% to 90% for the remainder of the melt

season. Pond depth increased steadily from a median depth of 0.40 m +/- 0.17 m in early June, peaked at 0.97 m +/- 0.51 m on 16 July, even as melt pond fraction had already started to decrease. Our results demonstrate that by combining high-resolution passive and active remote sensing we now have the ability to track evolving melt conditions and observe changes in the sea ice cover throughout the summer season.

## 1 Introduction

During the summer, highly reflective snow covered Arctic sea ice with an albedo > 0.7 decreases due to both the disintegration of the ice cover exposing the low-albedo open ocean (albedo < 0.1) and melt ponding on the ice surface (albedo 0.1 to 0.3) (Perovich and Polashenski, 2012; Light et al., 2022). This rapid change in albedo drives the positive ice albedo feedback (Curry et al., 1995), enabling additional uptake of shortwave radiation, enhancing melt. Meltwater percolation through the ice freshens the underlying ocean (Perovich et al., 2021), and further promotes ice disintegration and weakening of the ice

cover (Polashenski et al., 2012; Parkinson and Comiso, 2013), making it more vulnerable to breakup in summer storms. The





melt season concludes when freezing temperatures are sustained, the timing of which is geographically dependent. In mid-September the Arctic-wide ice cover reaches its lowest extent. The 44-year passive microwave record (1979-2022) reveals the September minimum extent is decreasing at a rate of -13% per decade (Fetterer et al., 2017) and this rate is accelerating (Comiso et al., 2008). The trend is -4.8% per decade from 1978-1996 and -14.9% per decade from 1997-2021 (Fetterer et al.,

2017). Markus et al. (2009) found the melt season lengthened at a rate of 6.4 days per decade from 1979 to 2007 based on the analysis of the timing of melt onset and freeze-up across the Arctic. Stammerjohn et al. (2012) also found a two month earlier retreat of the ice edge at the beginning of the melt season and one month later advance at the end of the melt season in regions where sea ice decrease is fastest (based on the 1979/80 to 2010/11 mean). Models predict an ice-free Arctic in late summer sometime this century (e.g. Wang and Overland, 2012; Arias et al., 2021). With observations of a declining summer sea ice

cover (Druckenmiller et al., 2021) and a lengthening of the summer melt season (Markus et al., 2009; Stammerjohn et al., 2012; Stroeve et al., 2014) it is essential that we better understand changes occurring throughout the summer on an Arctic-wide scale.

Sea ice melt processes have been studied during several dedicated field campaigns including the Surface Heat Budget of the Arctic Ocean (SHEBA) experiment in 1998 (Eicken et al., 2002; Perovich et al., 2002a, b, 2003), and during the

Multidisciplinary drifting Observatory for the Study of the Arctic Climate (MOSAiC) expedition in 2020 (Webster et al., 2022b), as well as through measurements on landfast ice near Utqiaġvik, Alaska (Perovich and Polashenski, 2012; Polashenski et al., 2012), and within the Canadian Archipelago (Yackel et al., 2000; Landy et al., 2014). Each of these studies describe stages of melt which we briefly summarize here: melt onset is geographically dependent but typically occurs in May or June (Markus et al., 2009). After the onset of melt, peak aerial coverage of melt ponds occurs lasting only a few days (Perovich and

Polashenski, 2012). During this time period, on level first year ice, melt water spreads across the smooth ice surface resulting in a maximum melt pond fraction as high as $\sim$ 50-70% (Grenfell and Perovich, 2004; Eicken et al., 2004; Polashenski et al., 2012), while on the rough topography of multiyear ice, lateral meltwater spread is prevented (Eicken et al., 2004; Petrich et al., 2012), resulting in a lower melt pond aerial fraction peaking at $\sim$ 30% (Fetterer and Untersteiner, 1998; Perovich et al., 2002b). Drainage channels form on the ice to efficiently route meltwater to either existing ponds, deepening them, or to channels that

run off ice floes (Eicken et al., 2002). Following the maximum pond fraction, the melt water can eventually drain through pores or macroscopic flaws that develop in the ice and ponds decrease in area. Ponds can melt through the sea ice and expose the open ocean, especially on thinner first year ice (Fetterer and Untersteiner, 1998; Eicken et al., 2002; Polashenski et al., 2012). At freeze onset, typically spanning mid-August to early September depending on location (Markus et al., 2009), the pond surface freezes, forming an ice lid that may accumulate snow (Flocco et al., 2015).

Remote sensing observations offer the potential to expand both the spatial and temporal scales over which summer melt can be studied. Tracking small-scale O(10 m$^2$) melt signatures from satellite platforms has proven challenging in the past due to limitations in resolution. Nevertheless, there have been successful observations of the evolution of local regions of sea ice using high resolution declassified governmental and commercial satellite imagery (e.g., Fetterer and Untersteiner, 1998; Kwok, 2014; Webster et al., 2022a, b; Niehaus et al., under review). The Moderate Resolution Imaging Spectrometer (MODIS)

(Rösel et al., 2012), Medium Resolution Imaging Spectrometer (Istomina et al., 2015), Landsat 7 Enhanced Thematic Mapper



(Markus et al., 2002, 2003), and synthetic aperture radar imagery (Mäkynen et al., 2014; Scharien et al., 2017), have all proven useful for studying melt ponds at a pan-Arctic scale, albeit at low resolution. Wright and Polashenski (2020) identifies the biases in the low resolution MODIS dataset and utilizes higher resolution, but spatially limited, WorldView imagery to improve the MODIS estimates of melt pond coverage. Several studies have explored the difference between ponding on first year ice and multiyear ice using both satellite observations (e.g., Fetterer and Untersteiner, 1998; Webster et al., 2015) and airborne observations (e.g., Buckley et al., 2020; Wright et al., 2020). Altimetric measurements from the Ice Cloud and land Elevation Satellite 2 (ICESat-2) have allowed for characterization of the altimeter's response to a melting surface (Tilling et al., 2020), and extraction of melt pond depth and width parameters (Farrell et al., 2020).

In situ and remote sensing observations have been essential for developing melt parameterizations in sea ice models (e.g., Flocco et al., 2010; Holland et al., 2012). However, the melt pond representation varies in complexity between parameterization schemes (Polashenski et al., 2012; Webster et al., 2022b). Some schemes employ a one-dimensional thermodynamical model to understand heat and mass transfer within the sea ice (Ebert and Curry, 1993; Taylor and Feltham, 2004), while others rely on the relationship between melt pond fraction and depth (Pedersen et al., 2009; Scott and Feltham, 2010; Hunke et al., 2013). Despite differences in melt pond parameterizations there is agreement that inclusion of melt processes in sea ice models significantly improves the prediction of end of summer sea ice thickness and extent (Flocco et al., 2010; Holland et al., 2012). However, while observations have served to improve our understanding of summer melt processes, data remain limited in time and space, leading to knowledge gaps (Webster et al., 2022b) and inadequate model parameterizations. For example, the evolution of pond fraction relative to sea ice type and the spatiotemporal variability in pond depth at Arctic-wide scales remain key unknowns (Webster et al., 2022b). Shu et al. (2020) found that although models included in the Coupled Model Intercomparison Project 6 (CMIP6) can capture the seasonal cycle of ice extent, most models overestimate the September minimum extent and there is still a broad spread across simulations, suggesting that sea ice melt processes are not well represented in models.

Now, new opportunities to detect and monitor melt ponds across the Arctic are available with the launch of earth observing satellites with high-resolution capabilities that also provide continuous measurements. This includes the ICESat-2, the first satellite laser altimeter to use photon counting technology (Markus et al., 2017). The ICESat-2 observational approach provides high-resolution surface height from which details of melt conditions on ice surfaces may be derived (Fricker et al., 2021; Farrell et al., 2020; Tilling et al., 2020). Evaluated alongside high-resolution visible and near-infrared satellite imagery, we can determine surface melt on Arctic sea ice and track its evolution. This study is motivated by the initial work conducted by Perovich et al. (2002b) and extends our understanding of the evolution of sea ice melt. Here, we use ICESat-2, Sentinel-2 and Maxar WorldView observations to derive sea ice concentration (SIC), melt pond fraction (MPF), and pond depth during the 2020 melt season. We describe two alternate approaches for tracking pond bathymetry and deriving depth from ICESat-2 observations. We present a timeline of melt evolution and explore the relationship between melt pond fraction and depth.





## 2 Study Period and Region

The 2020 annual mean surface air temperature across the Arctic was 2.1 °C above the 1981-2010 climatological mean, and warm temperature anomalies persisted from winter into summer across the Eurasian Arctic (Druckenmiller et al., 2021). As a result, the summer melt season of 2020 was an anomalous year of melt. May 2020 temperatures in the multiyear ice region were 1-5° C greater than average (Ballinger et al., 2020). In the Central Arctic, early melt onset occurred on 3 June 2020, and the date of continuous melt onset occurred on 16 June 2020, both dates six days earlier than the average for the time period 1979-2020 (Markus et al., 2009). September 2020 average sea ice extent was 3.92 million km$^2$, the second lowest on record (Fetterer et al., 2017). The 10-year merged CryoSat-2 - Soil Moisture and Ocean Salinity (CryoSat-2/SMOS) data record reveals an ice volume loss of 15,215 km$^3$ from April to October 2020, which resulted in the lowest recorded October ice volume (4,627 km$^3$) of the past decade (Perovich et al., 2020). We focus on the 2020 melt season because of these climate extremes, and analyze the evolution in melt conditions between 1 June and 15 September. Due to a satellite anomaly that resulted in extensive loss of Arctic sea ice observations in July 2019, 2020 also marked the first summer when continuous ICESat-2 records were available. The study thus begins prior to melt onset (Markus et al., 2009), and ends at the sea ice minimum as derived in the Sea Ice Index dataset (Fetterer et al., 2017), at which point optical imagery reveals refrozen leads.

The study region (Figure 1, purple shading), north of Greenland and the Canadian Arctic Archipelago extends from just west of Banks Island in the Beaufort Sea to northeastern Greenland, and includes the oldest and thickest ice in the Arctic (Bourke and Garrett, 1987). It was delineated from the multiyear ice extent on 15 May, 2020 prior to melt onset using a blended passive microwave and scatterometer sea ice type product provided by the EUMETSAT Ocean and Sea Ice Satellite Application Facility (Breivik et al., 2012). This was the latest-available observation of multiyear ice extent since the product is not available through the summer months due to the presence of surface meltwater that confounds the processing algorithm (Breivik et al., 2012). The study region is contained within the perennial ice area that persists at the end of the 2020 melt season (Comiso, 2002; Perovich et al., 2020) and overlaps with the "Last Ice Area" (Figure 1, gray shading), an area expected to retain multiyear ice in summer longer than any other part of the Arctic (Wang and Overland, 2009; Newton et al., 2021). We focus on this region since ice persists longest here in the summer and Farrell et al. (2020) have demonstrated the feasibility of retrieving melt pond depths on multiyear ice in the Lincoln Sea with ICESat-2 altimetry. Sentinel-2 imagery is widely available across the study region (Figure 1, pink dots) because of the proximity of multiyear ice to land (hence falling within the sampling mask used in Sentinel-2 acquisitions). Together with ICESat-2 elevation measurements, these observations provide a three-dimensional view of the sea ice surface.

## 3 Satellite Imagery

### 3.1 Sentinel-2 Observations

The Copernicus Sentinel-2 mission comprises two satellites, A and B, in a sun synchronous orbit each carrying the Multispectral Instrument (MSI) (Drusch et al., 2012) . The pair of satellites provide a global revisit time of less than 5 days. We use the



Level 1-C Top of Atmosphere products to derive parameters that describe changes in the ice cover throughout the summer. The
MSI samples across 13 spectral bands, four of which are used in our study: blue (B02, 492 nm), green (B03, 560 nm), red (B04,
665 nm), and near-infrared (B08, 833 nm). Data are provided at 10 m resolution. Sentinel-2 acquisitions are ideal for tracking
surface melt on Arctic multiyear ice since data are available for coastal waters within 20 km of the shore to a latitudinal limit of
84 °N (Drusch et al., 2012), as illustrated in Figure 1. So as to ensure high-quality surface observations, we required Sentinel-2
imagery with cloud-free areas exceeding 90%, the assessment of which was based on the Sentinel-2 cloud mask (Drusch et al.,
2012).

### 3.2 Maxar WorldView Observations

WorldView 2 and 3 provide higher resolution multispectral commercial satellite imagery, and are two of Maxar's WorldView
Legion. The satellites provide surface imagery across eight multispectral bands spanning 397 nm to 1039 nm, at 1.85 m and
1.24 m resolution, respectively. A set of 18 cloud-free images of summer melt with very high resolution (< 2m) are available
in our study region in 2019 and 2020 (Figure 1, black dots). WorldView images are processed and provided by the Polar
Geospatial Center (PGC) at the University of Minnesota. Here we analyze data from four spectral bands: blue (B02, 480 nm),
green (B03, 545 nm), red (B05, 645 nm), and near infrared (B07, 833 nm). Melt ponds on sea ice can range one to hundreds of
meters in diameter (Perovich et al., 2002b) which poses a challenge when using the Sentinel-2 imagery with 10-m resolution
for surface classification, such that there may be several surface types within a single Sentinel-2 pixel. WorldView imagery
has previously been used to study melt pond distribution and fraction in the Arctic (e.g., Lee et al., 2020; Li et al., 2020). The
higher resolution WorldView data are thus well suited for assessing the advantages and limitations of the Sentinel-2 data for
sea ice classification.

### 3.3 Image Classification

Image classification relies on the algorithm described in Buckley et al. (2020) that exploits natural breaks in the red, green,
and blue channel histograms to classify individual pixels as melt pond, sea ice, or open water. Prior to implementing this
classification procedure, we introduce a new step to distinguish water from ice by taking advantage of near-infrared obser-
vations provided in both the Sentinel-2 and WorldView multispectral data. Because water is very absorptive at near infrared
wavelengths (Curcio and Petty, 1951), data in the near-infrared channel can be used to discriminate between water and sea ice.
Following McFeeters (1996), we calculate the Normalized Difference Water Index (NDWI):

$$NDWI = (C_g - C_{NIR})/(C_g + C_{NIR}) \tag{1}$$

where $C_g$ is the green band (B03), and $C_{NIR}$ is the near-infrared band (B08 in Sentinel-2, B07 in WolrdView). NDWI is greater
for water than for ice surfaces due to the low reflectance of water at infrared wavelengths (McFeeters, 1996). In the NDWI
histogram, water pixels occupy the higher value bins. For unimodal histograms, a threshold ($H$) is set as the half maximum to
the left of the mode:

$$H = NDWI\_ma\_1\_hmL \tag{2}$$



If the NDWI histogram has more than one mode, we identify the mode with the highest pixel value ($NDWI\_ma\_m$) and in this case $H$ is the minimum ($mi$) to the left of $NDWI\_ma\_m$:

$$H = NDWI\_mi\_m \tag{3}$$

Pixels with $NDWI \leq H$ are non-water surfaces while those with $NDWI > H$ are classified as water pixels. Pixels classified
as water are subsequently further separated into either open water or melt pond pixels following the open water classification
approach of Buckley et al. (2020). All non-water pixels enter the sea ice classification step where they are classified as sea ice
or "other" pixels following the methodology described in Buckley et al. (2020). Pixels greater than the threshold (C) in the red
band ($C_r$) are identified as ice. Pixels less than the threshold, are identified as "other" pixels ($C_r <$ C, see Buckley et al., 2020).
"Other" pixels are those that are not as bright in $C_r$ as ice, and not as high in NDWI as water pixels. This includes mixed
pixels, pixels that include more than one surface type, and surface types such as newly-formed ice that is darker than the pixels
in the ice category. The same classification approach is used to analyze both the Sentinel-2 and WorldView 2 and 3 imagery.

We derived melt pond fraction (MPF), sea ice concentration (SIC) and open water fraction from the classification of indi-
vidual pixels. SIC is defined as the percentage of the sea surface that is covered in ice, open water fraction is the inverse; the
percentage of the sea surface not covered in sea ice. MPF is defined as the ponded percentage of sea ice (Buckley et al., 2020).
Figure 2 illustrates the classification scheme applied to multispectral imagery of sea ice in the Canada Basin as the surface
undergoes melt on 27 July, 2020. To quantify the limitations of the 10-m resolution Sentinel-2 dataset for observing small-
scale features such as melt ponds, we assess our SIC and MPF results in the context of a spatially and temporally coincident,
but higher-resolution, Maxar WorldView commercial satellite image. Here we show a near coincident WorldView-2 image for
comparison with the Sentinel-2 observation (12 minute time difference). We subset the Sentinel-2 image to the bounds of the
WorldView image applying the same classification algorithm as described above and compare results. In Figure 2c-f, we show
a segment of the WorldView and Sentinel-2 images and their classification masks. Figure 2e illustrates the high-resolution fea-
tures visible in the WorldView-2 imagery. In the center of Figure 2e, small melt ponds are connected by long, narrow drainage
channels. As these drainage channels are on the order of 5-10 m in width, the Sentinel-2 imagery does not resolve these fea-
tures, and pixels in this area are composed of both ice and meltwater (Figure 2c). Here, pixels consisting of small melt ponds
and drainage channels are classified as ice or "other" (Figure 2d). Figure 2 thus demonstrates the potential misclassification due
to the 10 m Sentinel-2 resolution. Other misclassifications include ridge shadows, submerged sea ice, very light melt ponds,
and open water, as previously discussed in Buckley et al. (2020) and in Section 3.4.

We estimate MPF of 7.6% and 25.5% from the Sentinel-2 image (Figure 2d) and WorldView (Figure 2f), respectively, a
difference of 18%. The underestimation of MPF (especially as the ice reaches the maximum MPF) in the lower resolution
image is consistent with previous studies (Buckley et al., 2020; Sivaraj et al., 2022; Niehaus et al., under review). We also
find SIC is 6.2% higher in the Sentinel-2 image than in the WorldView image. We further discuss the impact of the pixel
misclassifications and quantify the potential biases on the derived MPF and SIC in the next section.



## 3.4 Mixed Pixels and Pixel Misclassification

Small features on sea ice pose a challenge for satellite-derived classifications. Given ponds can range in size from less than 1
meter to 100s of meters in diameter (Perovich et al., 2002b), there may be several surfaces within a Sentinel-2 10-meter pixel.
Mixed pixels are those pixels with a combination of surface conditions, whether the edge of an ice floe, containing ice and open
water, or small melt ponds and drainage channels surrounded by sea ice. In these cases, it is difficult to robustly determine the
pixel designation since the reflectance signature is not indicative of one particular surface type. In our classification scheme,
these pixels are labelled "other." In Figure 2c-d the Sentinel-2 10 m pixels do not resolve the small melt pond features (Figure
2c, center), as seen in the WorldView image (Figure 2e, center). Pixel misclassification may occur in this situation where the
ponded sea ice is instead classified as sea ice (2d, center). Also, along the sea ice edge, where pixels contain both ice and
water, the pixels are classified as "other" (Figure 2d, green, center bottom). Also in the "other" category are complex ice types
such as new ice, which appears grey in imagery and is not bright enough to be classified as ice. This occurrence is rare, and
happens towards the end of the melt season as leads and areas of open water start to freeze. Pixels categorized as "other" are
not considered in the calculation of the derived parameters of MPF and SIC. Our analysis suggests that "other" pixels represent
on average less than 10% of all image pixels (see Section 3.1 and Figure 9, green).

## 3.5 Comparison of Satellite Image Classification

Given the biases revealed between the Sentinel-2 and WorldView analysis shown in Figure 2, we investigate the robustness of
the parameters derived during the Sentinel-2 classification. In Section 3.2 we discussed the ability to resolve small melt features
in WorldView imagery that are not resolvable in Sentinel-2 and showed an example in Figure 2. Our goal is to assess the level
to which MPF may be biased low due to the 10 m pixel resolution. We compare the MPF and SIC derived from Sentinel-2 with
the MPF and SIC derived from the higher-resolution WorldView imagery (Figure 3). We analyze 18 WorldView images that
were acquired during the 2019 and 2020 melt seasons and that are coincident with Sentinel-2 imagery. Although the imagery
spans two years, we organize the findings by day of year to understand if there is a seasonal trend in the bias. We identify
Sentinel-2 imagery captured within 24 hours of the WorldView imagery and we subsample the Sentinel-2 tiles to the extent of
the WorldView image by matching ice features in the imagery.

Comparison of the derived melt parameters from the classification of coincident images is shown in Figure 3 and Table 1.
In the beginning of the melt season, both data sets show consolidated ice with little or no signs of melt. The classification of
the images results in a good agreement in the derived MPF and SIC. In the five scenes in the first half of June, the MPF is less
than 1% in all Sentinel-2 and WorldView images. SIC is high in all the images (>90%), and the Sentinel-2 SIC agrees to within
3% of the coincident WorldView SIC (Figure 3 and Table 1). As the melt season progresses, sea ice floes are more susceptible
to break up due to structural weakening induced by melt pond formation (Arntsen et al., 2015) and enhanced dynamics as
sea ice is in free drift. For this reason, there are smaller features that appear in the imagery scenes: smaller floes, chunks of
ice, melt ponds and drainage channels. Small features are not as well resolved by the lower resolution of Sentinel-2 imagery
and thus misclassification and mixed pixels are more common. This leads to weaker agreement of the derived parameters in





Sentinel-2 versus those from the higher-resolution WorldView imagery, which still may be able to resolve these small features. The Sentinel-2 MPF is lower than MPF derived from WorldView images, as small ponds can go undetected or are classified as "other" pixels. From end of June through mid September, Sentinel-2 MPF is on average 12% lower, than the equivalent MPF derived from coincident WorldView imagery. There are two cases where the MPF calculated for the Sentinel-2 image is

greater than that of the WorldView image. In the imagery collected on 17 June 2020 the calculated MPF is 8.4% and 2.4% for the Sentinel-2 and WorldView images, respectively (Table 1). In this scene (see Figure 8b), there is level bare ice that appears blue in color, classified as melt pond in the Sentinel-2 imagery and ice in the Worldview imagery, resulting in a higher MPF for the Sentinel-2 scene than the WorldView scene. In the imagery captured on 3 September 2020 (WorldView subset shown in Figure 8f), there are many ice fragments smaller than Sentinel-2's pixel size (10 m) classified as "other" or melt ponds in the

Sentinel-2 imagery, falsely increasing the melt pond fraction. Our analysis shows that MPF can be biased low in the Sentinel-2 results by up to 20.7% and averaging 7.2%, when small ponds are widespread across the surface. SIC is biased high by up to 16% and averaging 4.3%, increasing as the melt season progresses (Table 1). The WorldView images better resolve these features and properly classify pixels as ice or open water.

In order to quantify the impact of pixel size on derived melt pond fraction, we look at the melt pond size distribution for each

WorldView image with a coincident Sentinel-2 image. With knowledge of the WorldView pixel size, we can determine the area of each object in the binary image. For each WorldView image, we determine the total area of ponds with a size smaller than the Sentinel-2 pixel area (100 m$^2$). The Sentinel-2 classification cannot resolve these small features as they are smaller than the pixel size. To quantify this, we calculate an adjusted Sentinel-2 MPF that adds the area unresolved melt ponds into the MPF calculation for each pair of coincident Sentinel-2 (S2) and WorldView (WV) images:

$$S2\_MPF_{adj} = \frac{S2\_MPA + WV_{100}\_MPA}{S2_{surf}\_A \times S2\_SIC} \tag{4}$$

where $S2MPFadj$ is the adjusted Sentinel-2 MPF, $S2\_MPA$ is the Sentinel-2 melt pond area, $WV_{100}\_MPA$ is the area of ponds less than 100 m$^2$ in size in the coincident WorldView image, $S2_{surf}\_A$ is the surface area in the Sentinel-2 image, and $S2\_SIC$ is the Sentinel-2 sea ice concentration. The denominator on the right-hand side is a calculation of the total area of sea ice and melt ponds in the Sentinel-2 scene. We make the assumption that melt ponds smaller than the Sentinel-2 pixel size

are classified as ice, so when making this adjustment, we hold the sea ice concentration constant and the area of WorldView melt ponds less than 100 m$^2$ replaces sea ice in the original Sentinel-2 classification. Table 1 provides the adjusted MPF results for each pair of WorldView and coincident Sentinel-2 images. Figure 3b shows the adjusted Sentinel-2 MPF as the sum of the royal blue and light blue bars.

Although the addition of ponds smaller than the Sentinel-2 pixel size through this adjustment increases the S2 MPF, making

it more comparable to the WorldView-derived MPF, it does not account for the entire discrepancy between the MPF derived from Sentinel-2 and WorldView (Figure 3b, Table 1). The average difference between the MPF derived from Sentinel-2 and WorldView decreased from 7.2% to 3.6% when the sub-pixel size WorldView ponds were accounted for. Although this methodology accounts for individual small ponds identified in WorldView imagery, we have not accounted for subpixel size areas that





are connected to larger ponds. Where the Sentinel-2 pixels may be classified as ice along the edges of ponds, a portion of that

pixel may be a melt pond, and properly classified as such in the WorldView classification. This scenario was not accounted for in the adjusted Sentinel-2 MPF, and may account for some of the remaining bias between Sentinel-2 and WorldView imagery. We also note that the Sentinel-2 derived SIC is on average 4.3% greater than that derived from WorldView, and with a higher SIC and sea ice area per scene, this contributes to a lower melt pond fraction.

## 4  Satellite Altimetry

### 4.1  ICESat-2 Observations

NASA's ICESat-2 satellite, launched in September 2018, carries a photon counting laser altimeter, the Advanced Topographic Laser Altimeter System (ATLAS), operating at 532 nm, with ground sampling every 0.7 m (Markus et al., 2017). ICESat-2 obtains surface height measurements across the Arctic up to 88°N with a 91-day repeat track orbit. The sole instrument onboard is the Advanced Topographic Laser Altimeter System (ATLAS), which has 3 beam pairs with 90-m spacing within

the pairs, and 3.3-km pair separation with the reference ground track (RGT) falling between the central beam pair. Each beam pair consists of a strong spot and a weak spot with an energy ratio of 4:1 (Neumann et al., 2019). We refer to the reference ground track (RGT) and beam as RGT yyyy GTNX, respectively, where yyyy is the track number, N is the beam pair number and X is L (left) or R (right) are the ground tracks (Herzfeld et al., 2021b). In this work, we exclusively use the strong beams to map sea ice topography and detect melt ponds. Previous studies have shown an elevation precision of 0.01 m can be achieved

over level sea ice surfaces (Farrell et al., 2020). The green laser is capable of penetrating clear water (Parrish et al., 2019), enabling measurements of shallow water-body depth including in nearshore bathymetry (Parrish et al., 2019; Babbel et al., 2021; Thomas et al., 2021), desert lakes (Armon et al., 2020), melt streams on ice shelves (Fricker et al., 2021), and sea ice melt ponds (Farrell et al., 2020). We use the ATL03 Geolocated Photon Height data product which provides photon height and geolocation above the WGS84 ellipsoid (Neumann et al., 2019, 2021), from which details of the sea ice surface and its

variability can be measured (Duncan and Farrell, 2022). Although Farrell et al. (2020) first demonstrated that the vertical resolution of ICESat-2 data is sufficient to resolve ponds on multiyear ice and manually estimated their depth, no operational ICESat-2 data product exists that automatically includes pond depth measurements. The higher-level ATL07 Sea Ice Height product (Kwok et al., 2021b) tracks sea ice surface height but does not have the ability to bifurcate and track two surfaces simultaneously, a requirement for pond depth retrievals.

### 4.2  Pond Depth Retrieval Algorithms

In this study, we use two unique algorithms specifically designed to track pond depths in the ICESat-2 photon cloud: the University of Maryland Melt Pond Algorithm (UMD-MPA) briefly described in Farrell et al. (2020), and the Density Dimension Algorithm (DDA) (Herzfeld et al., 2017, under review). Both algorithms operate on the ICESat-2 ATL03 geolocated photon height dataset to track the surface and bathymetry of individual ponds. We are able to estimate pond depth, an important





characteristic of melt ponds since it constrains meltwater volume and alters the hydrostatic balance of the sea ice (Webster et al., 2022b).

### 4.2.1 University of Maryland Melt Pond Algorithm

The UMD-MPA (Farrell et al., 2020) was developed to identify pond surfaces and their bathymetry in the ICESat-2 ATL03 photon height product (Neumann et al., 2021). First, we used a cloud indicator based on the apparent surface reflectance

parameter (Palm et al., 2021) provided as a flag in ATL07 (Kwok et al., 2021b), to identify cloud-free sections of along-track surface height data. If at least 20% of the track within the study region was cloud-free, we manually examined the ATL03 photon height data for evidence of melt ponds. Figure 4 demonstrates the methodology to determine the surface and bathymetry of a pond using the UMD-MPA. Figure 4a shows the ICESat-2 ATL03 photon cloud, where we see photons outlining the two-dimensional iconic bowl-shape of a melt pond (between 400 m and 700 m along track), with photons returned from both the

surface and bottom of the pond. We manually identified the start and end of ponds as the points where two surfaces diverge and rejoin, respectively. We defined pond width as the distance between the start and end points. To determine the surface height $h_s$, we binned all photons across the width of the pond into 0.1 m vertical bins (Figure 4b), and $h_s$ is the mode of the distribution:

$$h_s = P_n\_ma \tag{5}$$

where $P_n\_ma$ is the bin containing the maximum count in the vertically binned histogram for all photons across the width of the pond. $h_s$ was reset to an elevation of 0 m and all photon heights were recalculated relative to $h_s$ (Figure 4c). Then we constructed a new 2-dimensional histogram of photon height data with vertical elevation binned at 0.1 m using horizontal along-track bins 10 m wide in order to distinguish the surface photons from the bathymetric photons. For each vertical bin, we added the photons from the bins on either side to increase the photon count for each bin. In this way, the vertical bins were

overlapping with an effective bin height of 0.3 m at 0.1-m intervals (Figure 4d). For each 10-m horizontal along-track segment (as shown in Figure 4c), we examined the resulting histogram of vertical elevation (Figure 4d). We assumed photons within the two bins on either side of the identified pond surface mode could be associated with the surface and removed all photons in those bins for the subsurface analysis (Figure 4d, green bins), and thus the minimum retrievable pond depth was 0.3 m (0.23 m after correction for refraction of light in water). We located modes in the histogram below the surface that contained at least

5% of the number of surface photons in $P_n\_ma$ (Figure 4d, blue bin). If there were no modes that met this threshold, pond depth was not estimated at this location and we moved on to the next horizontal segment. If there were multiple modes, the one closest to the surface was defined as the bathymetry of the pond, as it was unlikely there are modes within a pond because the green laser is able to penetrate through the water column. The bathymetric elevation, $h_b$, of the pond was determined as the elevation of the subsurface mode:

$$h_b = P_{ni}\_ma_1 \tag{6}$$



Bathymetric elevation was determined for each 10 m horizontal section across the pond (Figure 4c, blue, Equation 6). Next, we estimated pond depth by differencing the pond surface and bathymetry. We then multiplied this depth by the ratio of the refractive index of air to water following Parrish et al. (2019) to derive the true melt pond depth $h_{mp}$, as follows:

$$h_{mp} = h_s - h_b \times \frac{\eta_a}{\eta_w} \tag{7}$$

where $h_{mp}$ is the depth of the melt pond, $h_s$ is the elevation of the pond surface, $h_b$ is the elevation of the bathymetry, $\eta_a$ is the refractive index of air (1.00029), and $\eta_w$ is the refractive index of water (1.33567) (Mobley, 1995). Pond depth ($h_{mp}$) was determined for each 10-m along-track segment. To increase along-track resolution, a linear interpolator with a 5-m length was applied to obtain pond depth at 5-m intervals across the pond. The true elevation within each bin could be +/- 0.15 m from the estimated value (half of the 0.3 m bin width). When the melt pond surface and bathymetry elevations are differenced to
determine the depth, uncertainty doubles because the pond surface and bathymetry uncertainties are additive (0.3 m), resulting in a total depth uncertainty of +/- 0.23 m after correction for refraction (0.3 x $\frac{\eta_a}{\eta_w}$). At least one depth measurement, and the melt pond start and end points are required for pond detection and thus the minimum retrievable pond width is 20 m. The advantage of the UMD-MPA is that individual ponds were manually selected so that false positives are minimized. However, the manual process of identifying ponds is arduous and vulnerable to human error.

**4.2.2   Density Dimension Algorithm – bifurcate – seaice**

The DDA is a family of fully automated algorithms designed to track complex surfaces in micro-pulse photon-counting lidar altimeter data, such as ICESat-2 (Herzfeld et al., 2017, 2021a, under review). The DDA-bifurcate-seaice algorithm was designed to track height in complex sea ice topography and has the ability to simultaneously track two diverging surfaces. A full description of the algorithm can be found in Herzfeld et al. (under review), but we briefly describe it here. The DDA-bifurcate-
seaice, hereinafter referred to as the DDA, utilizes the full geolocated photon height point cloud as provided in the ATL03 data product (Neumann et al., 2021). The algorithm employs the calculation of a density field for data aggregation and principles of auto-adaptive signal-to-noise thresholding and roughness determination (as described in Herzfeld et al. (2017)). The DDA has the ability to detect bifurcating reflectors, and can accommodate situations where the stronger reflector can be the lower or the higher reflector and the two reflectors may have different spatial distributions and material and reflection properties. The
DDA includes a layer follower with automated adaptation to layer roughness. On rough surfaces, the DDA tracks at 2.5-m intervals to capture the varying surface, and on smooth surfaces, at 5-m intervals. These parameters are adjustable. At least three sequential depth measurements are required for pond detection, and hence the minimum retrievable pond width is 7.5 m on a rough surface, and 15 m on a smooth surface. For comparison and consistency with the UMD-MPA, we resample the surfaces tracked by the DDA at 5-m intervals. The minimum elevation difference between the two tracked surfaces is adjustable within
the DDA, and for the purposes of this work it is set at 0.2 m within the photon cloud, allowing for a minimum retrievable pond depth of 0.15 m (after correction for refraction). The DDA is automated requiring no manual input and can be applied in





a systematic way. We use the DDA algorithm for comparison with the UMD-MPA, and to extend the time series of the melt pond depths in summer 2020.

### 4.2.3   Comparison of Pond Depth Retrieval Methods

Since the ATL07 algorithm (Kwok et al., 2021b) is designed to track only one sea ice surface height, the algorithms presented in this study are designed specifically to account for a melting sea ice surface and track two reflecting layers. Figure 5 shows three examples of melt ponds in ATL03 data and the performance of the UMD-MPA and DDA algorithms compared to the ATL07 surface tracking. Figure 5 a-c, (top panel) shows the ATL07 ICESat-2 product. Figure 5a shows that ATL07 tracks between the surface and the bathymetry of the two ponds, while in Figure 5b ATL07 tracks just the surface of the pond, and in

Figure 5c, ATL07 follows the bathymetry of the pond. This demonstrates the inconsistency of ATL07 tracking over a melted sea ice surface. The bottom panels of Figure 5a-c show the results of the UMD-MPA and the DDA tracking of the surface. Not only does this demonstrate the ability to track two surfaces, but also the consistent tracking, despite the differences in algorithm methodology.

We located 113 ponds that tracked by both algorithms and found a strong correlation between the mean pond depths (r=0.77,

Figure 11b). We found a mean residual difference of -0.04 m (DDA - UMD-MPA) with a standard deviation of 0.22 m (Herzfeld et al., under review). Although there is a small mean difference between the two algorithms, the standard deviation demonstrates some variability signifying remaining uncertainties tracking the location of the true melt pond bottom. Because of the good agreement between the two tracking algorithms, we combine the pond depths retrieved from both the UMD-MPA and DDA algorithms to analyze melt pond evolution throughout the summer (Section 5). Further discussion of the comparison of the two

algorithms is provided in Herzfeld et al. (under review).

### 4.2.4   Algorithm Limitations

The primary advantage of the DDA over the UMD-MPA is that it can be applied to any cloud-free ATL03 data over sea ice without user (manual) intervention. However, as a result of this automation, two scenarios associated with complex sea ice topography can result in false positive melt pond detection by the DDA. These cases are discussed in more detail in Herzfeld

et al. (under review), but here we illustrate two cases (Figure 6) and briefly describe our approach to reduce the impact of these issues. In the first case, complex surface topography associated with heavily deformed and ridged ice can result in the DDA algorithm tracking two surfaces between sea ice ridges (Figure 6a-b). Here, the surface tracking is not across a level pond surface, but instead the algorithm bifurcates and the first pass connects ridge sails and the second pass tracks rubble between the ridges. Anomalies such as these can be detected and discarded by flagging ponds that have surfaces with a standard deviation

> 0.05 m. This scenario has been eliminated with updates to the algorithm described in (Herzfeld et al., under review).

The second type of flase positive pond detection occurs due to the "dead time" of the ATLAS photon detectors when a strong surface return results in saturation of the detectors and a period of 3.2 ns where no additional photons can be detected (Smith et al., 2019; Lu et al., 2021). Following the detector "dead time", photons are once again reported resulting in a secondary "surface return" ~0.5 m below the true surface. Over sea ice surfaces, detector saturation commonly occurs over very bright



surfaces such as specular leads. In this scenario, the DDA tracks the secondary return as the bathymetry of a pond as seen in Figure 6c-d. To detect these occurrences, we look at the mean density of the surface return. The distribution of the mean density of the surface returns reveals a bimodal histogram. We have determined that the higher mode may correspond to a scenario where the surface is saturated and a secondary surface return results in false positives in the DDA tracking algorithm. Retracking anomalies due to dead time can be identified by depth measurements corresponding to the deadtime effect (0.5-0.6 m) where the surface mean density is greater than the minimum between the two modes in the mean density distribution.

### 4.2.5  Melt Pond Size Distribution

Using the 18 high-resolution WorldView images we calculate the melt pond size distribution. The goal of this analysis is to estimate the complete range of possible melt pond sizes on the surface and to determine which pond types are detected and profiled by the altimeter algorithms. For each WorldView image, we calculate the number of ponds, total pond area, mean pond perimeter, mean and median pond area, 5th and 95th percentile pond size, and mean circularity ($C$). Circularity ($C$) is measured for each individual pond following Perovich et al. (2002b):

$$C = \frac{P^2}{A} \tag{8}$$

where $P$ is the individual pond perimeter (m), and $A$ is the individual pond area (m$^2$). The minimum circularity (a circle) is $4\pi \sim 12.57$. The higher the circularity value, the more complex the pond perimeter. The results are tabulated in Table 2. Figure 7 shows the melt pond area distribution from the 18 WorldView images. We limit our analysis to melt ponds of at least 9 pixels (3 x 3 pixels), or 24.5 m$^2$ (30.8 m$^2$) in size, for WorldView 3 (WorldView 2), as smaller scales of melt ponds are indistinguishable from noise.

Although these WorldView images are not all from the same melt season or same location, we see patterns related to the stage of melt evolution. The monthly average pond area decreased from 227 m$^2$ in June to 163 m$^2$ in July and subsequently decreased in 156 m$^2$ in August to a low of 55 m$^2$ by early September. Pond perimeter averaged 71 m in June, decreased to an average of 64 m in July, increased slightly to 67 m in August, and finally decreased to a low of 38 m on average. The total pond area and number of ponds per image increases throughout the season until the end of August. In September both pond area and number of ponds per image decrease to the minimum value as freezeup occurs, as seen in the WorldView image on 3 September, 2020.

Mean pond circularity of all ponds in the WorldView images is 31.5 and ranges from 16.9 on 12 June to 40.7 on 9 August per image (Table 2). For comparison, a 1:7 rectangle has a circularity of 32.7. The high end of the range is similar to the value of 41.2 found on 7 August on multiyear ice in Perovich et al. (2002b). However, they found a mean pond circularity of 38.5 on 10 June. This difference could be due to our pixel-based algorithm detecting small melt ponds which tend to have a lower circularity. We find that the mean pond circularity per month increases as melt progresses: circularity averaged 29.9 in June, increased slightly to an average of 30.2 in July, increased slightly to 35.0 in August, and finally decreased to a low of 27.5 m on average in September. This indicates increasing pond complexity throughout the melt season.





Figure 7a demonstrates the prevalence of small ponds in the WorldView imagery. The the distributions from the three months show similar pond size distributions, but there is a slightly higher probability of larger ponds in July and August as compared to June, consistent with the findings of Perovich et al. (2002b). The cumulative probability distribution further illustrates the prevalence of small ponds, showing that 73% of ponds are smaller than 100 m$^2$ (Figure 7b). This implies that approximately 73% of individual ponds are not captured by the Sentinel-2 imagery which has a 100 m$^2$ pixel area (10 m pixel size). However, since these are small ponds, they account for only 38% of the total pond area in the WorldView scenes and Sentinel-2 imagery is able to capture approximately 62% of the total pond area. In Section 3.4 we discussed how the subpixel size melt ponds affect the Sentinel-2 derived melt pond fraction.

To estimate the approximate area of ponds with such widths, we assume a circular melt pond, resulting in the minimum detectable melt pond area is 314 m$^2$ and 44 m$^2$ for the UMD-MPA and DDA, respectively. Next we consider the minimum resolvable pond area when the UMD-MPA and DDA algorithms are used to map pond depths. Their capabilities are linked to the minimum pond width that can be detected which is 20 m for the UMD-MPA tracking (Section 4.2.1) and 7.5 m for the DDA tracking (Section 4.2.2). Figure 7b shows the cumulative distribution of individual melt pond area, with the minimum retrievable pond areas marked in magenta for the UMD-MPA and green for the DDA. Note that the cumulative distribution is shown for individual ponds, not of total ponded area. According to the WorldView imagery, approximately 83% of the total ponded area is made up of ponds with an area smaller than the UMD-MPA minimum resolvable size, suggesting the UMD-MPA is missing a large majority of ponds. However, the WorldView imagery melt pond distribution suggests that only 14% of the total ponded area is made up of ponds smaller than 44 m$^2$, the minimum detectable area for the DDA. The divergence in the UMD-MPA and DDA results are due to the inability of the UMD-MPA to track ponds smaller than 20 m in width, which makes up 96% of the melt ponds by number based on the WorldView imagery classification.

## 5 Results

The stages of melt pond evolution during summer 2020 from formation through freeze up are demonstrated in a time series of classified, high-resolution WorldView images (Figure 8). Figures 8a-f show WorldView RGB imagary and the surface classifications throughout the melt season. Figure 8g shows the evolution of SIC and MPF derived from each of the images in 8a-f. In the first image (Figure 8a), acquired on 9 June 2020, no ponds are visible on the ice surface (MPF = 0%, Figure 8g). At this point, the surface was melting and snow metamorphosing. By 17 June 2020 (Figure 8b), the meltwater had pooled into the lowest topographic areas, forming melt ponds (MPF = 3%, Figure 8g). By 30 June 2020 (Figure 8c), melt had advanced with a higher fraction of the ice covered in ponds (MPF = 23%). Drainage channels had formed between ponds by 22 July 2020 (Figure 8d) as ponds drained into other ponds and into the open ocean, either laterally or vertically and MPF is 25%. By 7 August 2020 (Figure 8e), regions of the ice had melted through, exposing the ocean. In some areas, the surface of the pond had refrozen to form an ice lid. These lids increased the albedo of the pond (Flocco et al., 2015), and restrict ICESat-2's laser penetration into the pond. Still, a large fraction of the ice was covered in ponds and the MPF peaked at 32%. In the image acquired on 3 September (Figure 8f), the majority of ponds had frozen ice lids that appear dark gray in color, and are classified




as ice. At this point, the MPF had decreased (MPF = 6%, Figure 8g). The refrozen leads are classified as "other" (green) in
       Figure 8f.

## 5.1    Summer Melt Parameters Derived from Satellite Observations

       We apply the classification algorithm described in Section 3.3 to 1,775 Sentinel-2 image tiles spanning the study region from
       1 June 2020 to 15 September 2020. The adjustments from Section 3.5 have not been applied as we do not have coincident
WorldView imagery for all the Sentinel-2 tiles and calculate MPF and SIC. MPF is calculated for images with SIC > 15% so
       as to reduce the pixel misclassifications associated with mixed pixels at the sea ice edge and brash ice. We look at images in
       a running 15-day period and identify images with anomalously high melt pond fraction (>95th percentile). Anomalously high
       MPF was identified in 79 images (4% of total tile count). Of these, 75% (59 tiles) were either contaminated with clouds that
       evaded the initial cloud masking procedure (Drusch et al., 2012), or included the presence of fast ice. These tiles were discarded.
The remaining 25% (20 tiles) were determined to be uncontaminated and properly classified, and retained for analysis.

### 5.1.1    Feature Classification

       We examine the evolution of surface classifications throughout the melt season (Figure 9). Vertical gray bars indicate signal
       loss due to the requirement of 90% cloud free images (Section 3.1) when there are fewer than ten images in the five-day period.
       At the beginning of the melt season a high percentage of pixels (>90%) are classified as ice, and this is followed by a sharp
drop to <80% in mid-June. The ice pixel percentage decreases through mid-August and then becomes more variable. Melt pond
       pixels increase from 3.0% on 13 June to 10.0% on 15 June and the maximum coverage is 15.3% on 24 June. The percentage of
       pixels classified as melt ponds remains greater than 10% until July 2, and makes up less than 5% of each image from July 23
       through the end of the study period (15 September). The open water percentage is low (<5%) at the beginning of the season and
       then increases and becomes more variable later in the season, with the highest open water percentage from mid-July through
mid-August. This indicates an increase in lateral melting of floes and a more dynamic, divergent ice cover. On average, open
       water makes up 14% of the surface pixels in July and 17% in August. The open water percentage decreases in late August and
       September as leads begin to refreeze. Throughout the season, the pixels classified as "other" remain below 10%. Towards the
       end of the season, refrozen leads appear in the Sentinel-2 scenes and the algorithm classifies these areas as "other", explaining
       the increase in "other" pixel percentages in September (Figure 9).

### 5.1.2    Sea Ice Concentration

       We examine the SIC derived from Sentinel-2 data in the study region. Mean SIC in the region was 91.6% with a standard
       deviation of 15.0% and the median was 97.2%. The difference between the median and mean indicate that there are some
       Sentinel-2 tiles with very low SIC, or entirely open water. The SIC values ranged from 0-100%, with 75% of the SIC values
       greater than 92.8% and 25% greater than 99.0%. As the melt season progressed, individual images had more variable SIC
and the median SIC value decreased. Figure 10 shows the seasonal evolution of the melt parameters with the SIC shown in





Figure 10a. SIC was consistently greater than 90% through mid-June. On 27-28 June, imagery show the ice separated from the landfast ice in the Lincoln Sea and along the western coasts of the Canadian Archipelago (Vermote and Wolfe, 2021). At the same time, sea ice drift data indicate westward ice drift (OSI-SAF, 2022). These dynamics opened leads and reduced local ice concentration. Throughout July, the sea ice continued to separate from the coast leaving large areas of open water. As the ice cover receded, Sentinel-2 images along the edge of the pack ice captured lower SIC (<80%) and in the lowest latitudes of the study region, SIC values dropped below 20% (Figure 10a). The consolidated ice cover evolved into a mosaic of smaller floes with leads that grew in size as the floes melt laterally. Median SIC dropped below 80% in late July, consistent with Perovich et al. (2002b), who observed a sharp decrease from 95% to 80% SIC in early August in aerial observations of the SHEBA site.

### 5.1.3  Melt Pond Fraction

We calculated MPF from the Sentinel-2 images in the study region with SIC >15% (Figure 10b). The average MPF in the region in summer 2020 was 6.5% with a standard deviation of 6.5%. The highest MPF for an individual Sentinel-2 scene was 31.6%. This image is located just outside of the mouth of Nansen Sound at -95.3° W, 82.3° N, but far enough from that coastline that it does not contain landfast ice. Median MPF remains low, <5%, through 17 June. We then see a sharp increase to 12.1% in MPF on 18 June. The imagery is scarce between 18 and 22 June due to widespread cloud coverage. This weather system likely enhanced the melt (Mortin et al., 2016), and when it passed, the MPF was high, averaging 15.2% between 24 and 29 June. The peak five-day running mean MPF was 15.9% on 24 June (Figure 10b). The MPF slowly decreased in July and by August, the MPF remained below 5% for the remainder of the season. The evolution of melt in WorldView images, presented in Figure 8f, follows a similar pattern: a sharp increase in MPF earlier in the season and a decrease in MPF by September. However, the images show a sustained high MPF (>20%) through early August (Figure 8f), indicative of the variability of MPF regionally and at smaller scales.

### 5.2  Melt Pond Depth

Of the 1107 ICESat-2 tracks that traversed the study region between 1 June 2020 and 15 September 2020, only 850 tracks met the cloud cover requirements described in Section 2.3.1. Upon examination of the ATL03 data acquired along these tracks, we identified 477 individual melt ponds (Figure 11a). The UMD-MPA was applied to these ponds resulting in over 11,000 individual pond depth measurements. We applied the DDA to 87 of 850 (10% of the available ICESat-2 tracks in the study region and period) cloud free tracks that are representative in time and space of the study region throughout the melt season. We do not apply the DDA to the central beam (beam GT2L) as the central beam is more susceptible to specular returns and the dead time effect in the summer (Kwok et al., 2021a). For the DDA post-processing, we discard all anomalies associated with the heavily deformed/ridged sea ice and those arising due to the detector dead time effects (Figure 6) using the approaches described in Section 4.2.3. This process discards 5,319 of 94,543 individual pond measurements, corresponding to 5.6% of the available measurements. The DDA tracked 7,329 ponds with a total of 89,224 individual depth measurements after the post processing steps (Figure 10c, Figure 11a).





For each individual pond we find the median pond depth (Figure 10c). Herzfeld et al. (under review) demonstrates the strong agreement between the median depths of ponds tracked by both the DDA and UMD-MPA (r = 0.77, mean different 0.04

m), so we combine the two datasaets (Figure 10c). The DDA median pond depth evolution (not shown) is very close to the evolution of the full dataset (UMD-MPA and DDA combined) because there are many more ponds tracked by the DDA than the UMD-MPA. The melt pond depth evolution, Figure 10c, is not representative of a single pond, but the evolution of the parameters of all ponds in the study region. Individual ponds have complex meltwater accumulation and vertical and lateral drainage processes, which are not captured in the evolution of the entire study region presented in Figure 10. Throughout the

season, we see a widening of the interquartile range (IQR), suggesting ponds across the area were in different stages of melt. Freeze onset in August caused the ponds to form ice lids, preventing laser penetration into the pond for pond depth retrieval, although there may have been liquid water beneath the ice lid. Freeze conditions occur at different points in this region at the end of summer, and there are fewer pond depth measurements throughout the month of August.

Although ponds were observed in Sentinel-2 imagery in early June, the first melt pond depth measurements from the UMD-

MPA are on 22 June 2020. This indicates that the ponds present early in the season were shallow ponds and ICESat-2 measurements of any individual pond did not exceed the minimum retrievable pond depth (0.23 m), and thus pond depth was not retrieved. The DDA has the ability to track smaller, shallower ponds, where the UMD-MPA relies on manual identification of ponds that biases the results towards larger ponds.

## 6 Discussion

Factors controlling melt progression include end of winter snow depth, ice topography, solar radiation, latitude, and weather events (Eicken et al., 2004). In this section we discuss our results in context of existing literature, understanding that melt pond evolution varies based on seasonal surface conditions and regional atmospheric events.

### 6.1 Evolution of Sea Ice Conditions

#### 6.1.1 Early Melt

From 1 June through 17 June, MPF was less than 5% and SIC greater than 99% (Figure 10a-b). Figure 8 shows an example of an unponded ice surface on 9 June (Figure 8a) and early melt occurred in the image observed on 17 June (Figure 8b). During this time the median DDA-derived depth remained below 0.5 m and there were no ponds tracked by the UMD-MPA. At the MOSAiC site in the same melt season as our study but in the Fram Strait east of our study region, continuous melt started in mid-June 2020 expanding existing ponds and increasing the pond areal coverage (Webster et al., 2022b).

#### 6.1.2 Maximum Melt

Our results show a sharp increase in MPF in mid-June (Figure 10b), consistent with Perovich et al. (2002b) where aerial observations over the field site show a rapid increase in pond fraction over the study area from 5% to 20% in just a few days.





Scott and Feltham (2010) also find a sharp increase in the modeled MPF early in the melt season in their standard multiyear case. We found MPF greater than 10% from June 23 through July 2, with a maximum MPF of 16% on 24 June. At the MOSAiC

site on primarily second year ice, ponds greater than 100 m in diameter were observed on 1 July 2020 (Webster et al., 2022b). Maximum pond coverage occurred later in the season at SHEBA in 1998 (24% on 7 August) (Perovich et al., 2002b) and at the MOSAiC site in 2020 (21% on 26 July) (Webster et al., 2022b). In the second half of June, the first ponds were tracked by the UMD-MPA (22 June), and both UMD-MPA and DDA median pond depths increased through the end of June. Similarly, at MOSAiC, melt pond depths increased through early July. Scott and Feltham (2010) found gradually increasing pond depth

in their model. Morassutti and Ledrew (1996) analyzed 220 pond depth measurements on multiyear ice within the Canadian Archipelago from 27 May to 26 June 1994. These ponds had a mean depth of 0.27 m with a standard deviation of 0.13 m. The UMD-MPA measurements revealed a higher mean melt pond depth for this time period, (0.75 m +/- 0.66 m), but this is likely due to the part of the melt pond size distribution sampled by ICESat-2 and the UMD-MPA minimum observable depth of 0.23 m. From 4-26 June 2020, the DDA tracked ponds had a mean pond depth of 0.54 m with a standard deviation of 0.4 m.

### 6.1.3 Late Season Evolution

Following the maximum MPF on 24 June, there was a decrease in MPF, consistent with Eicken et al. (2002) and Polashenski et al. (2012), who both described decrease in pond coverage as meltwater was efficiently routed through drainage channels, ice permeability increased, and meltwater percolated through the sea ice. At MOSAiC, a drainage event that occurred in mid-July reduced the pond area by 5% (Webster et al., 2022b). Eicken et al. (2002) and Perovich et al. (2002b) suggest a second mode

of MPF as melt ponds spread laterally and connect through drainage channels, but we did not see this in our results. This could be a result of the low resolution of the Sentinel-2 imagery where the smaller drainage channels that occurred later in the melt season were not resolved well in the imagery and melt pond pixels were classified as ice pixels instead. The UMD-MPA median pond depth increased throughout July, from 0.32 m on 1 July to 0.93 on 30 July, whereas the DDA depth increased through 16 July, reached a median depth of 0.78, then was less than 0.5 m from 19 July through 14 August. These contrasting

results demonstrate the bias of the UMD-MPA toward identification of larger melt ponds, and inability to track smaller ponds. However, evolution is highly dependent on local weather and sea ice conditions (Webster et al., 2022b), and it is likely that the UMD-MPA and DDA were tracking ponds under different atmospheric and sea ice conditions. The simulated pond depth in Scott and Feltham (2010) surpassed 1 m in early July, and remained above 1 m for the remainder of the melt season, agreeing well with the UMD-MPA observations. The gradual increase in melt pond depth throughout the season was also observed at

SHEBA (Perovich et al., 2003). Observations at MOSAiC show that pond depth increased over time and melted through the first year ice by late July (Webster et al., 2022b).

### 6.1.4 Refreeze

The formation of ice lids on melt ponds is a sudden process, as below freezing temperatures will quickly freeze the top layer of the pond, drastically reducing pond fraction (Webster et al., 2022b). Small, shallow ponds form lids before larger, deeper ponds

(Webster et al., 2022b). The MOSAiC observatory was relocated to the central Arctic (approximately at 89 °N) in mid-August





and although MPF was greater than 30% on 4 September, all ponds had refrozen by 6 September 2020, effectively reducing the MPF to 0%. The number of melt ponds tracked by the UMD-MPA and DDA (gray histogram in 9c), significantly decreased toward the end of August indicating ponds had either drained or a lid had formed preventing laser penetration into the pond. This is consistent with our findings of low MPF during this period. The WorldView image observed on September 3 (Figure 8e) shows light gray ponds, indicative that a pond lids had formed.

## 6.2 Relationship Between Pond Depth and Fraction

We consider our MPF and depth evolution results in context of the existing depth-area relationship used to parameterize ponds in the Community Earth System Model (CESM) and level-ice formulation available in the Community Ice CodE (CICE) (Holland et al., 2012; Hunke et al., 2013):

$$h_p = 0.8 \times MPF \tag{9}$$

Where MPF is melt pond fraction as a percent (%) and $h_p$ is pond depth in centimeters.

The ratio is based on a time series of depth and fraction observations from a 200-m albedo line at SHEBA in 1998. The SHEBA observations reveal a constant linear relationship between pond fraction and pond depth (Perovich et al., 2003). However, Polashenski et al. (2012) show that in their study over landfast ice in northern Alaska in 2009, the relationship between the pond fraction and depth cannot be described by any function. Similarly, there was no clear relationship observed between pond fraction and depth at the MOSAiC field campaign (Webster et al., 2022b); MPF increased as the depth increased until early July, then the MPF increased but the mean pond depth remained fairly constant. In this study, the median melt pond depth and MPF increase through June, but as MPF began to decrease, the depth continued to increase (Figure 12). Our results suggest that there is no simple relationship between pond depth and fraction, but nevertheless we hope these findings can provide insight into how pond depth and fraction evolve.

The study presented here shows the feasibility of conducting such analyses over large regions of the ice cover. More work is needed to understand the evolution of these parameters at both local scales and Arctic-wide. We have only applied the DDA to a small subset of available ICESat-2 tracks, and further analysis may provide additional information to better characterize the relationship.

## 6.3 Limitations

Satellite measurements of summer sea ice provide a time series of Arctic-wide observations, a scale unobtainable from in situ and airborne studies. However, resolution is limited, introducing errors and biases in satellite-derived products. Comparisons with higher-resolution WorldView images suggests that MPF estimates derived from Sentinel-2 are biased low by 7.2% on average, and up to 20% at the peak of the melt season (Sections 4.2.5 and 5.1.3). The bias can be quantified and corrected using higher resolution WorldView imagery when available (Section 4.2.5). The pond depth retrieval (Section 4.2.4), is limited by the ICESat-2 laser pulse width (0.2 m) (Neumann et al., 2019; Tilling et al., 2020). Further, the methodology for the UMD-MPA requires manual identification of ponds which favors larger ponds, restricting pond depth estimates to the largest ponds.



The UMD-MPA and DDA tracking results show good agreement (Figure 5), (Herzfeld et al., under review) and the datasets are combined to increase sampling for analysis. Still, 14% of the ponded area is not sampled due to the minimum width requirement of 7.5 m for the DDA. Despite these limitations, this study demonstrates the ability to track small scale features of summer sea ice over long time periods and large areas from satellites.

## 7   Summary and Conclusions

Arctic sea ice conditions in summer 2020 were anomalous with above average May surface temperatures, a near record-setting end of September ice extent, and record ice volume loss over the melt season (Druckenmiller et al., 2021). Using new, high-resolution remote sensing observations, we tracked changes in melt pond fraction and depth across perennial sea ice. We adapted algorithms developed in previous work (Herzfeld et al., 2017; Buckley et al., 2020; Farrell et al., 2020) to analyze a larger dataset and provided new details about the evolution of melting sea ice conditions during the 2020 melt season. Melt pond fractions increased from melt onset until June 24, peaked at $\sim 16\%$ and then decreased for the remainder of the summer with variability between the Sentinel-2 scenes. These results were consistent with previous studies conducted on multiyear ice that showed rapid MPF increase in mid-June (Perovich et al., 2002b), and maximum MPF at the end of June (Rösel and Kaleschke, 2012). The combined results from the two depth-tracking algorithms applied to ICESat-2 revealed that median and mean pond depth remain below 0.50 m until mid-June when they slowly increased through July. The evolution of melt pond depth is consistent with previous studies (Section 6.1). The minimum retrievable depth of 0.23 m with the UMD-MPA and the manual identification of ponds favoring large ponds, resulted in the derived depths from the UMD-MPA bias high compared to previous studies (Morassutti and Ledrew, 1996; Perovich et al., 2003; Webster et al., 2022b).

The UMD-MPA is not an automated algorithm and requires manual inputs on pond locations, and thus cannot be employed for operational processing of ICESat-2 products. On the other hand, the DDA has the ability to automatically track multiple surfaces in situations of complex spatial data distributions and mathematically difficult signal to noise ratios. In this study, we demonstrated the ability of the DDA to track ponds on multiyear ice but only on a subset (10%) of the available data. The automated DDA can be applied to all summer sea ice tracks to efficiently extract important melt pond information. With a higher density of pond observations spread through time and space, we will be able to analyze these observations at multiple scales, and better understand spatial and temporal patterns.

While we have demonstrated the ability to derive melt parameters from the region of thick, predominantly multiyear ice, there is potential to extend the investigations of summer melt by including ICESat-2 and Sentinel-2 observations over seasonal ice. However, tracking ponds on first year ice presents additional challenges. Ponds on thin ice are shallower and melt through the ice faster than they would on multiyear ice (Morassutti and Ledrew, 1996). Our ability to track shallow ponds is limited by ICESat-2's 0.2 m pulse width (Neumann et al., 2019; Tilling et al., 2020). Sentinel-2 data over first year ice is also limited because during the summer, the first year ice area retreats off the coast of western Canada and Alaska, and imagery is only available within 20 km from the coast. Despite these challenges, it is important to study the evolution of melt ponds on first



year ice, as it is the dominant ice type in the Arctic. We suggest further development of algorithms that can systematically be applied to summer ICESat-2 ATL03 data to track melt ponds.

These findings can be put in context of the in situ and airborne measurements conducted as part of the MOSAiC campaign during this same time period (Shupe et al., 2020). Although the study region here did not overlap with the MOSAiC drift locations, there may be similar patterns in the evolution of melt parameters. The ICESat-2 measurements of melt pond depth

presented in this study will benefit from in situ and airborne validation campaigns. Dedicated in situ campaigns are required for better understanding the melting sea ice surface and structure of the complex pond bottom. Airborne measurements of melt ponds, with coincident or near-coincident ICESat-2 passes, can further validate the melt pond depth retrievals, and quantify the uncertainty from ICESat-2 measurements over the melting sea ice surface. For example, in July 2022, NASA conducted an airborne validation campaign to survey perennial ice north of Greenland. Six flights mapped sea ice beneath coincident

ICESat-2 orbits and these data will be used for assessment of the accuracy of ICESat-2 observations of summer sea ice.

The melt parameters derived in this study may be useful for advancing parameterization of melt ponds in sea ice models. These products can enhance our understanding of the under-ice light and biology (Horvat et al., 2020; Light et al., 2008), as light transmission through melt ponds penetrates to the upper ocean during summer (Light et al., 2008), stimulating biological activity (Arrigo et al., 2012). Pond depth and area measurements provide a three-dimensional view of surface ponding and is

valuable for quantifying the volume of meltwater stored on perennial ice (Zhang et al., 2018). Melt ponds reduce the overall albedo of sea ice (Fetterer and Untersteiner, 1998; Perovich and Polashenski, 2012), and meltwater drainage affects the freshwater budget of the upper ocean (Perovich et al., 2021). Pond volume can also be used to estimate how the presence of ponds alters the hydrostatic balance assumed when deriving sea ice thickness from altimeter measurements of sea ice freeboard. This study demonstrates the feasibility of using high resolution remote sensing observations to understand summer sea ice evolution.

*Code and data availability.* The image classification algorithm is available at https://github.com/ellenbuckley/MeltEvolution. The image classification results and melt pond depth database is archived on Zenodo, DOI: 10.5281/zenodo.7568995. ICESat-2 ATL03 data are available at https://nsidc.org/data/atl03, ATL07 data are available at https://nsidc.org/data/atl07, Sentinel-2 data were downloaded from the Sci-Hub: https://scihub.copernicus.eu/ using the SentinelSat API: https://sentinelsat.readthedocs.io/, WorldView imagery is courtesy of the Polar Geospatial Center, the OSI SAF Global Sea Ice Type product is available at https://osi-saf.eumetsat.int/products/osi-403-d

*Author contributions.* SLF, KD and EB conceived of the study. The UMD-MPA was developed by EB with assistance from SLF and KD. UH developed and executed the DDA with assistance from TMT, ML and HH. EB acquired and processed the Sentinel-2, WorldView and ICESat-2 data with support from OB and KD. Data analysis was conducted by EB and SLF with support from all authors. TMT, ML, and HH processed the region-wide data sets of ATL03 ICESat-2 data using the DDA. MW provided feedback throughout the project. EB and SLF wrote the paper, with contributions from MW and UCH. EB performed this work at the University of Maryland. All authors reviewed

the manuscript. We acknowledge the support of Jaemin Eun for help organizing the classification workflow into publicly-releasable code.



*Competing interests.* The authors declare they have no competing interests.

*Acknowledgements.* WorldView geospatial support for this work provided by the Polar Geospatial Center under NSF-OPP awards 1043681 and 1559691. This study is supported under NASA Cryosphere Program Grants 80NSSC17K0006, 80NSSC20K0966 and 80NSSC22K0815 (PI: Farrell) and by NASA's Earth Sciences Division under awards 80NSSC20K0975, 80NSSC22K1155, 80NSSC18K1439 and NNX17AG75G (PI: Herzfeld). MW acknowledges support from NASA's New Investigator Program in Earth Science (80NSSC20K0658) and the National Science Foundation (2138786).




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



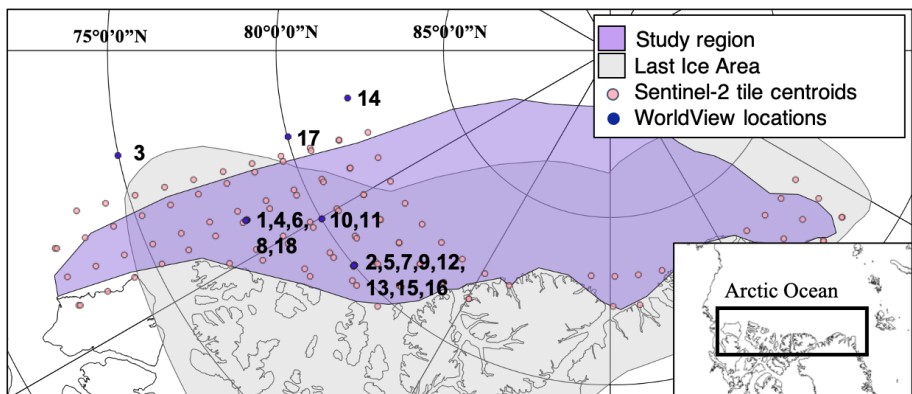

**Figure 1.** The study region (purple shading) north of Greenland and the Canadian Arctic Archipelago (inset) is based on the location of multi-year ice in May 2020 and intersects the last ice area (gray shading). Sentinel-2 tile centroids (pink dots) indicate availability of satellite image acquisitions. Centroids of a subset of WorldView imagery (black dots) are numbered.





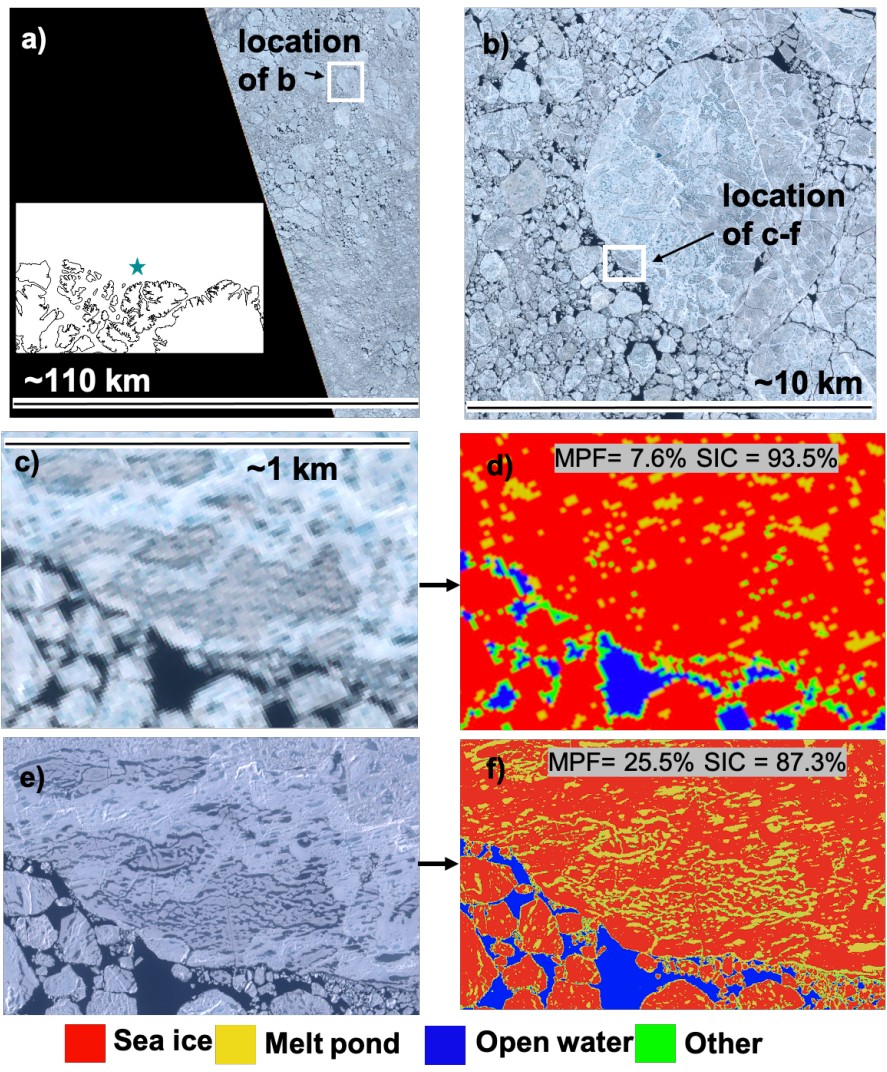

**Figure 2.** Classification of satellite images of sea ice at 80° N, -110° W acquired on 27 July, 2020. (a) True-color Sentinel-2 image. (b) A 10 km x 10 km subset of the Sentinel-2 image outlined by the white box in (a) showing circular ice floes of different sizes. An area of sea ice 1 km x 0.8 km in size outlined in white shows the location of images (c)-(f). (c) 1 km x 0.8 km subset of the Sentinel-2 image illustrating ice floes undergoing surface melt. (d) Classification of image pixels in (c) showing sea ice (red), melt ponds (yellow), open water (blue), and other pixels (green). (e) WorldView image of sea ice that is spatially and temporally coincident with (c) (tile 13 in Figure 1). (f) Classification of image pixels in (e), color-coding same as in (d). Melt pond fraction (MPF) and sea ice concentration (SIC) derived from classified data, in units of %. (WorldView imagery copyright 2020 Maxar).





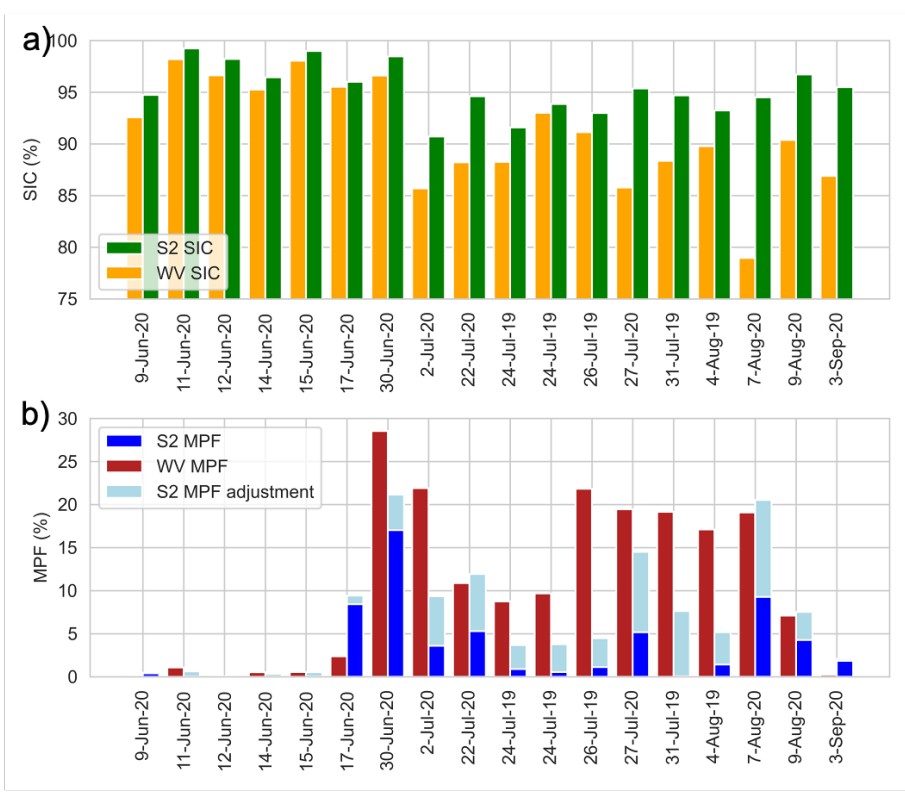

**Figure 3.** Comparison of derived melt parameters in coincident WorldView and Sentinel-2 images. a) SIC Sentinel-2 (green) and WorldView (gold). b) WorldView images (red), MPF Sentinel-2 (blue) and the adjusted Sentinel-2 MPF (shown as sum of light blue and blue bar).

**Figure 4.** Schematic demonstrating the UMD-MPA methodology. a) ATL03 photon height cloud (grey dots) revealing a melt pond located in the center of the transect. b) Histogram of photon heights spanning 1 km along track and binned at 0.1 m vertically. The primary mode indicates the surface (black). c) A 300-m long section across the pond in a) and the horizontal binning at 10 m intervals. The yellow box marks the horizontal section analyzed in the vertical histogram shown in d). In d) the surface bin and two bins on either side are green and the subsurface mode in blue. e) Melt pond surface (black dots), bathymetry (magenta dots), and corrected depth (gray bars) derived using UMD-MPA applied to the ATL03 data.



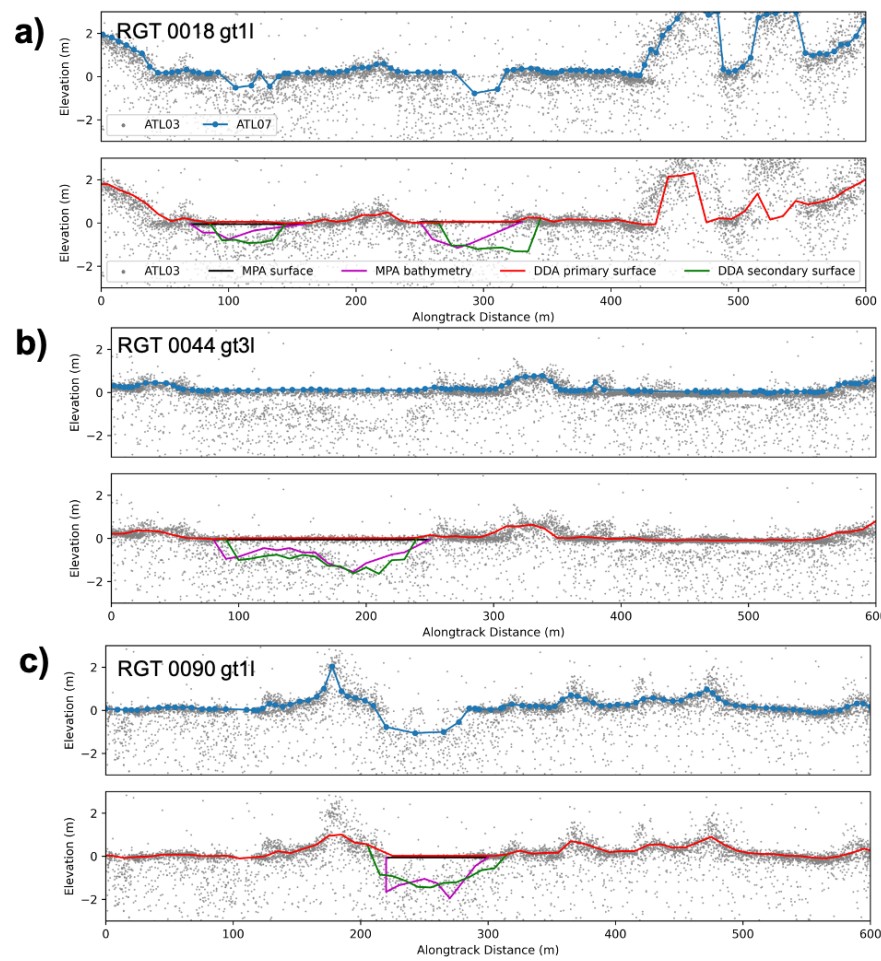

**Figure 5.** Surface tracking algorithms over ponded sea ice surfaces. Panels show surface height results from three algorithms applied to the ATL03 photon height data: ATL07 (blue), DDA-bifurcate-seaice primary surface (red), DDA-bifurcate-seaice secondary surface (green), UMD-MPA surface (black) and UMD-MPA bathymetry (magenta), along ICESat-2 reference ground tracks (RGT) (a) 0018, (b) 0044 and (c) 0090.



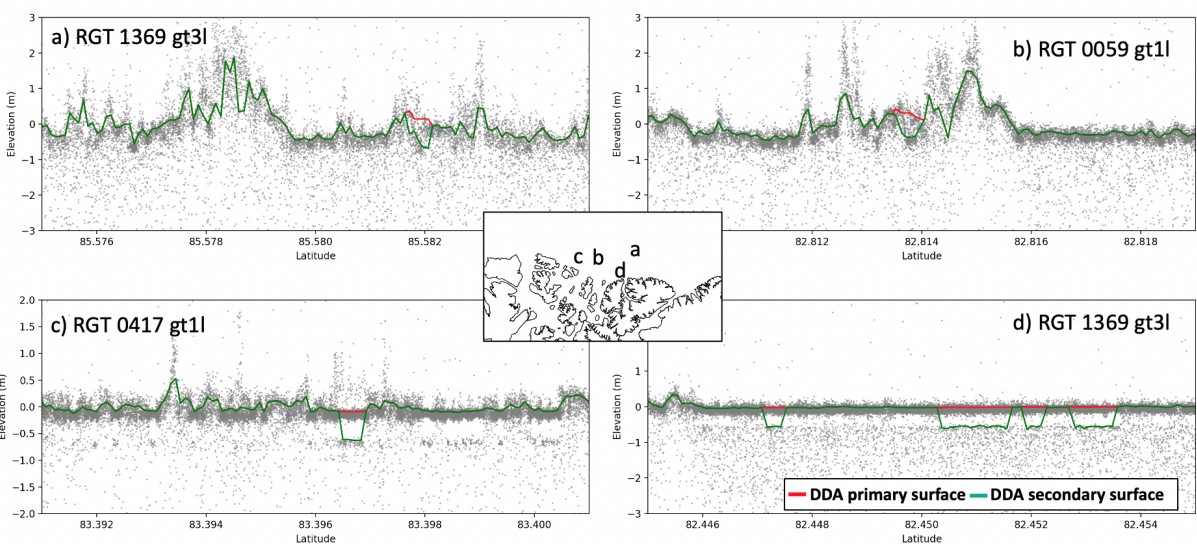

**Figure 6.** Anomalous melt pond detections from the automated DDA-bifurcate-seaice tracking algorithm. Examples (a) and (b) show the result of DDA bifurcation in regions of heavily deformed ice, where the surface height of ice blocks scattered across a rubble field are tracked as the primary surface (red) and the height of the consolidated ice is the secondary surface (green). Examples (c) and (d) the subsurface deadtime effect. The inset maps the location of the four surfaces shown in a) through d).



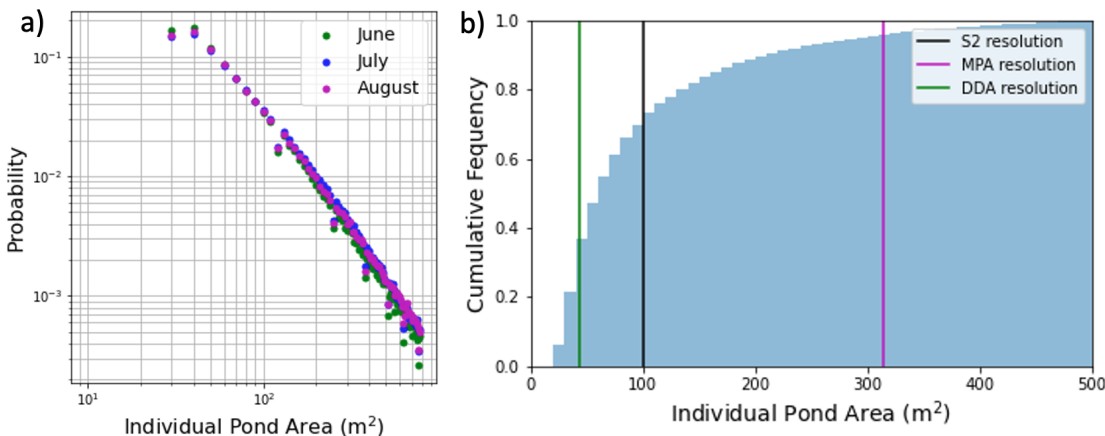

**Figure 7.** Melt pond size distribution calculated from 2019 and 2020 WorldView imagery. a) Melt pond area distribution colored by month: June (green), July (blue) and August (magenta). Area bins of size 10 $m^2$ were used. b) cumulative individual melt pond area distribution. The Sentinel-2 individual melt pond area resolution (100 $m^2$) is shown as a solid black line. The melt pond area corresponding the minimum resolvable UMD-MPA and DDA-bifurcate-seaice widths (20 m, 7.5 m, respectively) and assuming circular melt ponds are shown in magenta and green, respectively.



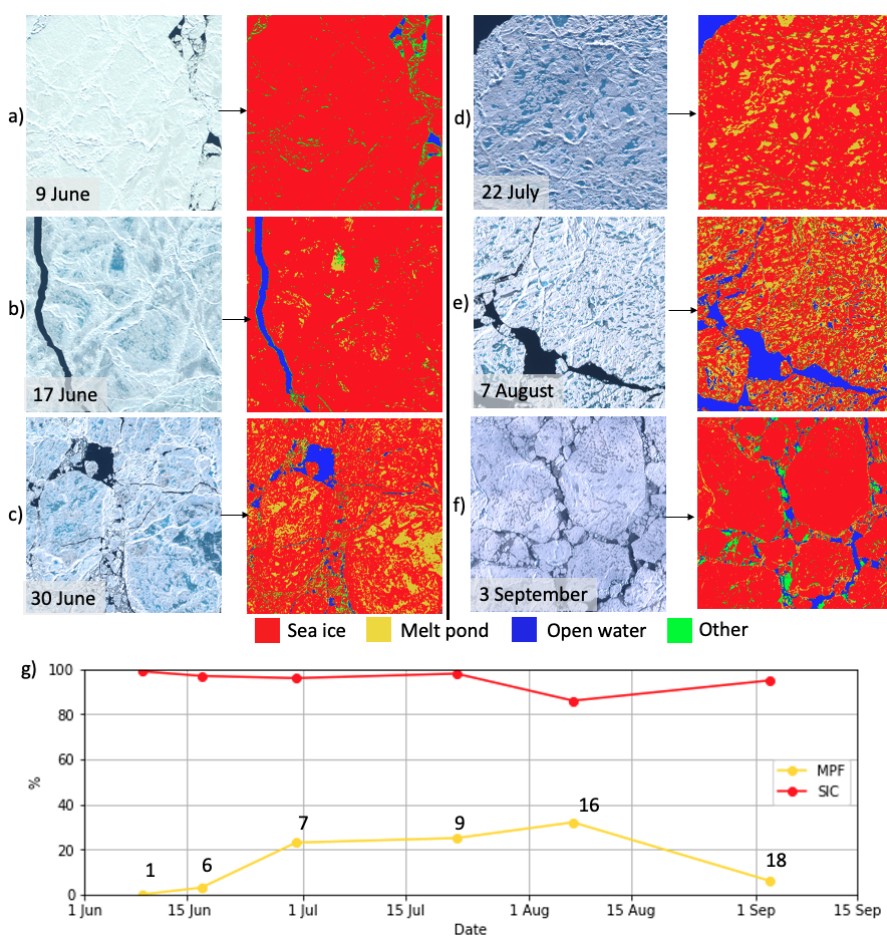

**Figure 8.** Melt evolution in 2020, based on a selection of WorldView imagery (∼ 900 m x 900 m in area) spanning 9 June - 3 September, 2020. Figures (a)-(f) show the RGB true color composite (left) and the classified image (right). g) MPF (gold) and SIC (red) derived for each image. These images are from two different locations within the study region, the corresponding image numbers in g) mark their location in Figure 1 with more information in Table 1. (WorldView imagery copyright 2020 Maxar).



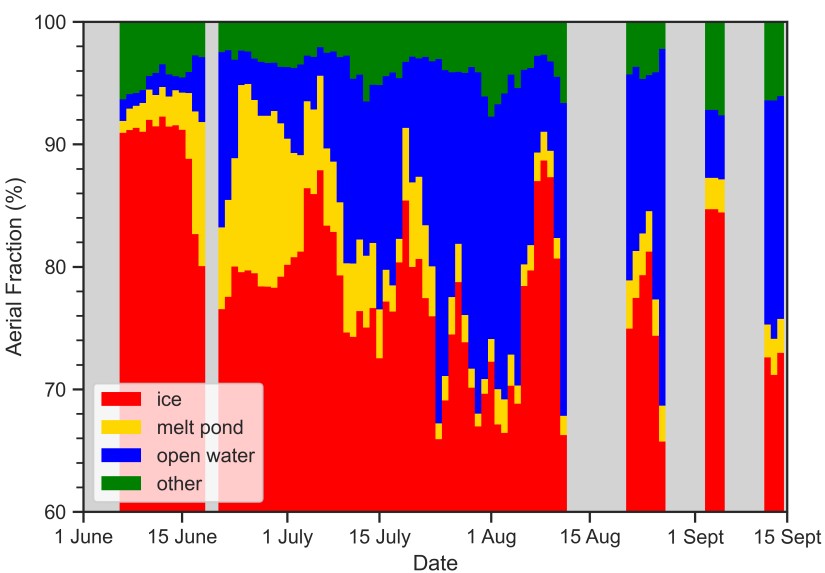

**Figure 9.** The five-day mean aerial fraction of surface types from classification of Sentinel-2 imagery throughout the 2020 Arctic melt season. Surface pixels are classified as ice (red), melt pond (yellow), open water (blue), or "other" (green). The gray bars indicate that there are fewer than ten images in the five-day period.



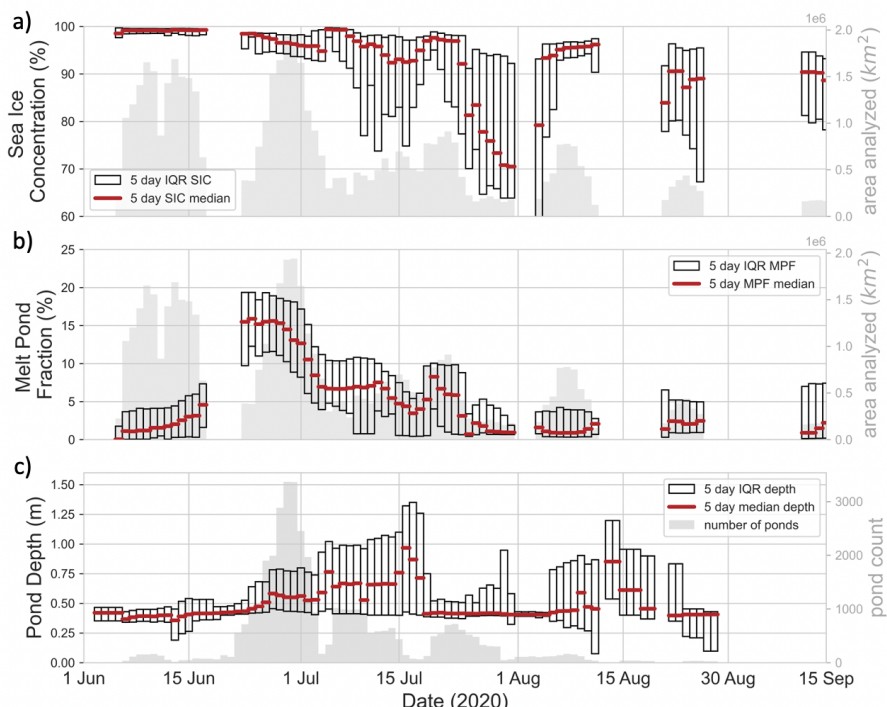

**Figure 10.** Evolution of melt features from 1 June 2020 to 15 September 2020 in the study region. a) box plot showing the median sea ice concentration for a 5 day window centered on the plotted date. The box shows the interquartile range. The gray bar plot in background shows the total area of Sentinel-2 imagery analyzed per five-day window. b) same as in a) but for melt pond fraction. c) same as in a) and b) but the median pond depth from merged DDA-bifurcate-seaice and UMD-MPA tracked ponds for a 5 day window centered on the plotted date.



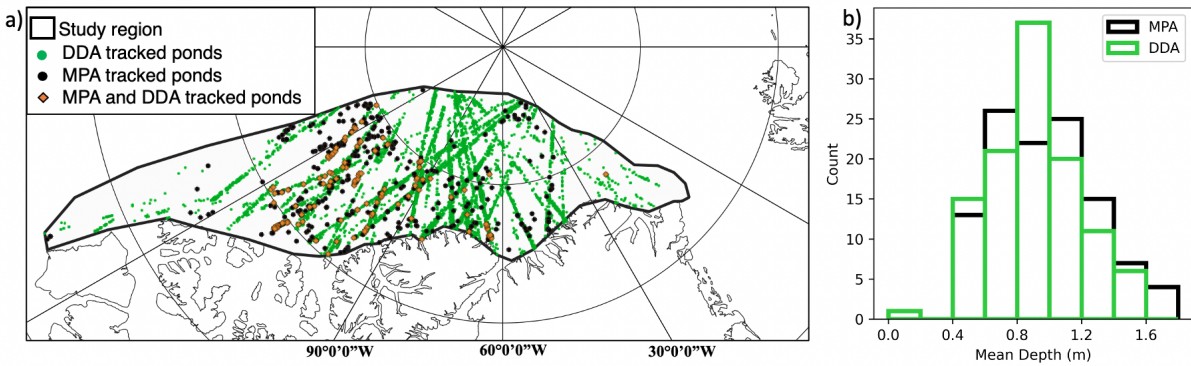

**Figure 11.** Melt ponds measured by both algorithms. a) locations of the ponds measured with the DDA-bifurcate-seaice (green circles), UMD-MPA (black circles), and both algorithms (orange diamonds) in the study region (black outline). b) mean depth of melt ponds measured by both algorithms: DDA-bifurcate-seaice (green), MPA (black)



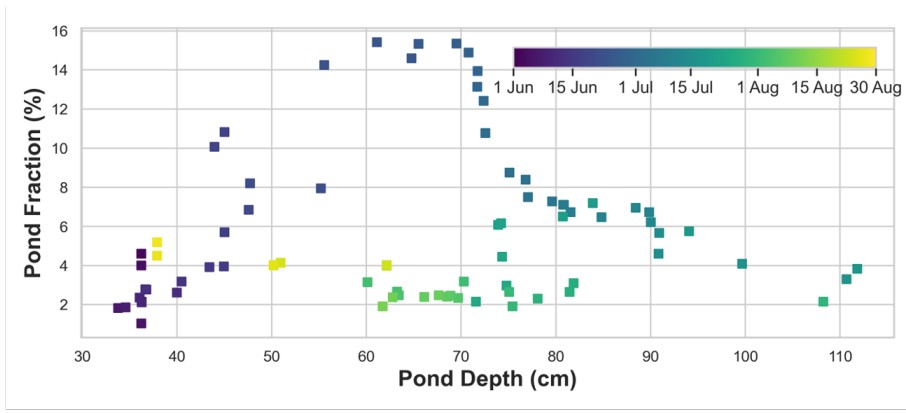

**Figure 12.** Relationship between observed pond fraction and pond depth during the multiyear ice region in the 2020 melt season, colored by time. Five-day median pond depth and fraction are shown.





**Table 1.** Derived melt pond fraction (MPF) and sea ice concentration (SIC) from coincident WorldView and Sentinel-2 images.

| Image Number | Date | Central Latitude | Central Longitude | S2 SIC (%) | WV SIC (%) | Δ SIC (S2-WV) (%) | S2 MPF (%) | WV MPF (%) | Δ MPF (S2-WV) (%) | S2 MPF adj (%) | Δ MPF adj (S2 - WV) (%) |
|---|---|---|---|---|---|---|---|---|---|---|---|
| 1 | 9-Jun-20 | 78.0 | -125.0 | 94.7 | 92.6 | 2.2 | 0.4 | 0.1 | 0.3 | 0.5 | 0.3 |
| 2 | 11-Jun-20 | 80.0 | -110.0 | 99.2 | 98.2 | 1.1 | 0.0 | 1.1 | -1.1 | 0.6 | -0.5 |
| 3 | 12-Jun-20 | 75.0 | -138.0 | 98.2 | 96.6 | 1.6 | 0.0 | 0.1 | -0.1 | 0.1 | 0.0 |
| 4 | 14-Jun-20 | 78.0 | -125.0 | 96.4 | 95.3 | 1.2 | 0.0 | 0.5 | -0.5 | 0.3 | -0.2 |
| 5 | 15-Jun-20 | 80.0 | -110.0 | 99.0 | 98.0 | 0.9 | 0.0 | 0.5 | -0.5 | 0.5 | 0.0 |
| 6 | 17-Jun-20 | 78.0 | -125.0 | 96.0 | 95.5 | 0.5 | 8.4 | 2.4 | 6.0 | 9.4 | 7.0 |
| 7 | 30-Jun-20 | 80.0 | -110.0 | 98.5 | 96.6 | 1.9 | 17.0 | 28.5 | -11.5 | 21.1 | -7.4 |
| 8 | 2-Jul-20 | 78.0 | -125.0 | 90.7 | 85.7 | 5.0 | 3.6 | 21.9 | -18.3 | 9.4 | -12.6 |
| 9 | 22-Jul-20 | 80.0 | -110.0 | 94.6 | 88.2 | 6.4 | 5.3 | 10.9 | -5.6 | 11.9 | 1.0 |
| 10 | 24-Jul-19 | 80.0 | -119.8 | 91.6 | 88.3 | 3.3 | 0.9 | 8.8 | -7.9 | 3.7 | -5.1 |
| 11 | 24-Jul-19 | 80.0 | -119.8 | 93.9 | 93.0 | 0.9 | 0.5 | 9.7 | -9.1 | 3.8 | -5.9 |
| 12 | 26-Jul-19 | 80.0 | -110.0 | 93.0 | 91.1 | 1.8 | 1.1 | 21.8 | -20.7 | 4.5 | -17.4 |
| 13 | 27-Jul-20 | 80.0 | -110.0 | 95.4 | 85.8 | 9.6 | 5.2 | 19.4 | -14.3 | 14.5 | -5.0 |
| 14 | 31-Jul-19 | 82.0 | -139.7 | 94.7 | 88.4 | 6.3 | 0.0 | 19.1 | -19.1 | 7.6 | -11.5 |
| 15 | 4-Aug-19 | 80.0 | -110.0 | 93.2 | 89.8 | 3.5 | 1.4 | 17.1 | -15.7 | 5.2 | -11.9 |
| 16 | 7-Aug-20 | 80.0 | -110.0 | 94.5 | 79.0 | 15.6 | 9.3 | 19.1 | -9.8 | 20.5 | 1.5 |
| 17 | 9-Aug-20 | 80.0 | -135.0 | 96.7 | 90.4 | 6.3 | 4.3 | 7.1 | -2.8 | 7.5 | 0.4 |
| 18 | 3-Sep-20 | 78.0 | -125.0 | 95.5 | 86.9 | 8.6 | 1.9 | 0.2 | 1.6 | 2.1 | 1.8 |



**Table 2.** Melt Pond Area Distribution derived from WorldView Imagery

| Date | number of ponds | total melt pond area (m²) | mean melt pond perimeter, $\overline{P}$ (m) | mean melt pond area, $\overline{A}$ (m²) | 5th percentile melt pond area (m²) | median melt pond area (m²) | 95th percentile melt pond area (m²) | mean melt pond circularity, $\overline{C}$ |
|---|---|---|---|---|---|---|---|---|
| 9-Jun-20 | 972 | 58110.6 | 37.4 | 59.8 | 34.2 | 47.9 | 124.8 | 24.7 |
| 11-Jun-20 | 13644 | 1318542.1 | 55.1 | 96.6 | 34.2 | 58.2 | 277.2 | 33.8 |
| 12-Jun-20 | 877 | 66143.2 | 34.2 | 75.4 | 34.2 | 51.3 | 172.5 | 16.9 |
| 14-Jun-20 | 6249 | 423921.1 | 36.3 | 67.8 | 34.2 | 51.3 | 160.9 | 20.9 |
| 15-Jun-20 | 10130 | 816281.7 | 43.0 | 80.6 | 27.2 | 49.0 | 231.4 | 26.3 |
| 17-Jun-20 | 34839 | 3451221.6 | 43.8 | 99.1 | 34.2 | 61.6 | 284.1 | 21.0 |
| 30-Jun-20 | 168405 | 47213983.0 | 81.5 | 280.4 | 34.2 | 71.9 | 633.2 | 32.1 |
| 2-Jul-20 | 177058 | 34206970.3 | 62.3 | 193.2 | 34.2 | 68.5 | 441.5 | 26.4 |
| 22-Jul-20 | 36733 | 3594007.6 | 49.3 | 97.8 | 27.2 | 54.4 | 304.9 | 28.1 |
| 24-Jul-19 | 119124 | 16611644.5 | 56.5 | 139.4 | 34.2 | 75.3 | 438.1 | 25.5 |
| 24-Jul-19 | 95524 | 12228537.7 | 50.9 | 128.0 | 34.2 | 71.9 | 362.8 | 23.6 |
| 26-Jul-19 | 155825 | 39487319.6 | 81.2 | 253.4 | 34.2 | 75.3 | 742.7 | 30.4 |
| 27-Jul-20 | 304298 | 38099628.8 | 61.3 | 125.2 | 27.2 | 57.2 | 397.5 | 34.8 |
| 31-Jul-19 | 236057 | 39536174.9 | 67.6 | 167.5 | 27.2 | 79.0 | 536.3 | 32.2 |
| 4-Aug-19 | 164640 | 29114310.8 | 73.3 | 176.8 | 34.2 | 75.3 | 585.2 | 34.1 |
| 7-Aug-20 | 232460 | 29996627.5 | 67.4 | 129.0 | 27.2 | 57.2 | 400.2 | 40.7 |
| 9-Aug-20 | 122476 | 22072722.4 | 56.5 | 180.2 | 34.2 | 65.0 | 400.4 | 25.4 |
| 3-Sep-20 | 785 | 42863.4 | 38.0 | 54.6 | 34.2 | 44.5 | 112.9 | 27.5 |