# Peer review of "Observing the Evolution of Summer Melt on Multiyear Sea Ice with ICESat-2 and Sentinel-2"

_EGUsphere, 2023_

## Referee Comment (RC1)

Review of

**Observing the Evolution of Summer Melt on Multiyear Sea Ice
with ICESat-2 and Sentinel-2**
by Buckley et al.

https://doi.org/10.5194/egusphere-2023-189

**General comments:**

The paper presents high-resolution data of melt pond properties in three dimensions (melt pond fraction, size distribution, depth), which are obtained from satellites (Sentinel-2, Maxar WorldView, ICESat-2) of the newest generation. Analysis is focused on the region of most persistent ice cover in the Arctic, where second- and multiyear-ice is the dominant type of sea ice. The data set covers the whole 2020 melt season, which allows to investigate the full life-cycle of melt ponds from formation to refreeze.

Melt pond depth is retrieved from ICESat-2 by using two different algorithms. UMD-MPA requires manual input, DDA is an automated algorithm. The comparison of the results of both algorithms is very valuable as well as the description of the limits like minimum retrievable pond width and depth. The systematic analysis of pond depth in a wider region of the Arctic Ocean is a big step forward to better understand the three-dimensional evolution of melt ponds during the melt season. Such data is also extremely valuable for validation and improvement of melt pond parameterisations in sea ice models. One example, the relation between pond depth and melt pond fraction, is given in the present paper.

As cloud-free conditions are a prerequisite for retrieving melt pond properties from the three satellites used, it is still not possible to analyse melt pond evolution in a very high resolution in time in a wider area. However, airborne campaigns could bridge certain gaps. As outlined in the conclusions, such airborne observations as well as in situ observations (coincident with ICESat-2 passes) will also be very helpful to quantify the uncertainty from ICESat-2 measurements.

To summarize, the paper addresses very relevant scientific questions within the scope of TC, and it presents novel data. Substantial conclusions are reached. The scientific methods are clearly outlined and the results are sufficient to support the conclusions.

The title reflects the content of the paper, the abstract provides a complete summary and the paper is generally well structured. The review of existing published work is very good, the number of references is appropriate. Overall, figures and tables are clear and their captions self-explanatory. Mathematical formulae, symbols and abbreviations are correctly defined and used. The use of the English language is very good.

**Specific comments:**

It might be worth to add another aspect to the discussion. Based on the described errors and limitations in the melt pond properties retrieved from Sentinel-2 and ICESat-2, what is the impact of those errors and limitations on estimates of surface albedo?

**Technical corrections:**

Line 93/94: Please, check the years in "September 2020 average sea ice extent … (Fetterer et al., 2017)"

Line 395: Please., check "decreased in".

Line 407: "The the distributions …", delete "the".

---

## Referee Comment (RC2)

**Review of Buckley et al. Observing the Evolution of Summer Melt on Multiyear Sea Ice with ICESat-2 and Sentinel-2**

The authors provided a mainly methodological paper that integrates various remote sensing sources to track the evolution of melt pond characteristics (melt pond fraction and depth) on multiyear sea ice in the perennial ice zone north of the Canadian Arctic Archipelago in the 2020 melt season. This region contains some of the oldest and thickest ice in the Arctic and is distinct from the broader and largely first-year ice dominated Arctic pack. Multispectral optical data from Sentinel-2 and Worldview-2/3 satellites are used to estimate melt pond fraction (MPF) by image classification, and the resolution limitations of Sentinel-2 (10m) are evaluated in the context of the classifier by using the high-resolution Worldview-2/3 images (~1m) as verification. This is useful and insightful, as it provides new information about the limitations of the widely available but resolution constrained Sentinel-2 to estimate and study MPF broadly. As perhaps expected, the Sentinel-2 classifier underestimates MPF due to its inability to resolve smaller scale features such as small ponds and interconnecting meltwater channels due to pixel mixing. Despite being expected, it is good to see this problem being addressed rigorously and the limitations quantified. The authors found Sentinel-2 MPF is biased low "by up to 20.7% and averaging 7.2%, when small ponds are widespread across the surface". The authors also provide a method for adjusting the Sentinel-2 MPF based on input Worldview imagery when available. This approach is limited by the availability of high-resolution optical imagery, which should be less of a problem in the future as more open access data of that type becomes available.

The authors further compare two novel melt pond depth retrieval algorithms that use ICESat-2 altimetry data as input (ATL03 geolocated photon height) and discriminate pond surfaces from bottoms to resolve pond depth. They are unique, such that a comparison like this provides detail essential to their effective application to studies of atmosphere-ice-ocean processes during advanced stages of sea ice melt, when melt ponds play a key role in energy and meltwater budgets, among other things. The limitations of the algorithms are explicitly and thoroughly addressed in the context of the detection limits of ICESat-2, e.g. the pulse width imposes a 0.2m limit on pond depth, which means depth would not likely be resolved on seasonal sea ice due to their shallow nature. As the authors clearly indicate, one of the algorithms (DDA) is superior to the other (UMD-MPA) for detecting pond depth in a larger proportion of melt ponds in the studied area because it can resolve smaller ponds.

Sea ice and melt pond fractions are combined with melt pond depth estimates to examine the evolution of the advanced melt season in 2020 and the evolving parameters are discussed in the context of known key periods (early melt, maximum melt, etc.), previous observations from field campaigns SHEBA and MOSAiC, and model parameterizations (CESM and CICE). In general, the authors draw similarity between their results and those of others except when comparison to the modelling approaches where there is no observed relationship between pond depth and fraction.

The paper is original and presents new insights on the combined evolution of melt pond fraction and depth, thus providing an important third dimension to the study of these important features using remote sensing data. The methods are valid and suitable and connections to parallel and

ongoing work on the melt pond depth algorithm (DDA) are clearly and helpfully stated. Enough detail is provided to understand both application potential and limitations of the described methods. The topic is definitely appropriate for *The Cryosphere*, though the authors should first address the below major and minor comments.

**Major comments**

The paper is imbalanced in that it presents as an analysis of the 2020 melt season due to its relevance in a climate context and in terms of the sea ice record – anomalously warm spring, early melt, and the second lowest September sea ice extent on record. It is more focused on the methods associated with melt pond fraction and depth retrieval and limitations. By comparison, the melt season analysis is rather cursory and mostly limited to a time series analysis of the derived parameters. The authors could strengthen the paper by either placing more emphasis on the methodological components as the core theme of the paper (e.g. section 5.2.4 belongs in results), or by expanding the melt season analysis so that the 2020 melt season evolution, as it is characterized by the retrieved melt pond properties, is better understood in the context of being an anomalous season. The latter would benefit from a comparison of 2020 conditions to other years which understandably would be limited to 2021 and 2022 by Sentinel-2 and ICESat-2 availability.

The authors should better describe the combined and relative importance of melt pond fraction and depth for the study of sea ice melt season evolution and atmosphere-ice-ocean interactions. This could be more explicitly addressed through the stated research goal(s) of the paper. Outlining the potential benefits of merging depth estimates with pond fraction, i.e. for volume estimates, and for improved understanding of melt evolution, is needed. How tracking the depth in only the larger ponds would influence the results is also of relevance. This is needed partly due to the imbalanced nature of the paper, as mentioned above. Detail regarding the importance of these observations mainly comes at the end of the conclusions, whereas the emphasis is on the methods early on and only general statements are made (e.g. line 83, understanding the evolution of melt).

Minor Comments (L=line)

L90: Clarify what area is the multiyear ice region.

L120: Give the spectral range as done for Worldview below.

L146: "Worldview"

L168: Delete Maxar

L173: Pixel misclassification is discussed in section 3.3 and 3.4 and both discussions refer to the same figure. Should be in one section.

L178: Add "image" after Worldview.

213: Use more precise terminology than chunks.

L218: Delete second comma

L229: Use MPF instead of melt pond fraction.

L233: area "of" …

L238: Use SIC after Sentinel-2.

L253: MPF

L259: Use ATLAS only, it was defined earlier.

L371: "false"

L379: Does the dead time effect also happen for leads as it does for melt ponds?

L402: The Perovich et al. 2002b paper is cited, but a little more context about the study would still be appropriate.

L406: How do we know that this is due to seasonal evolution versus differences in the area covered by the Worldview scenes and the ice topography etc.?

L429: "imagery"

L436: Provide detail on how the presence of ice lids is determined. Is it the dark grey color mentioned, in which case can this be confused with a drained melt pond?

L555: Can delete ", and inability to track smaller ponds"

L614: "…being biased high…"

Figure 6 caption: word missing "Examples (c) and (d) the subsurface". Show?

---

## Author Comment (AC1)

Review of
Observing the Evolution of Summer Melt on Multiyear Sea Ice with ICESat-2 and Sentinel-2
by Buckley et al.
https://doi.org/10.5194/egusphere-2023-189

Thank you for the review of the manuscript. Your suggestions and comments have improved the manuscript. The following responses (in blue font) address your comments point by point.

General comments:
The paper presents high-resolution data of melt pond properties in three dimensions (melt pond fraction, size distribution, depth), which are obtained from satellites (Sentinel-2, Maxar WorldView, ICESat-2) of the newest generation. Analysis is focused on the region of most persistent ice cover in the Arctic, where second- and multiyear-ice is the dominant type of sea ice. The data set covers the whole 2020 melt season, which allows to investigate the full life-cycle of melt ponds from formation to refreeze.
Melt pond depth is retrieved from ICESat-2 by using two different algorithms. UMD-MPA requires manual input, DDA is an automated algorithm. The comparison of the results of both algorithms is very valuable as well as the description of the limits like minimum retrievable pond width and depth. The systematic analysis of pond depth in a wider region of the Arctic Ocean is a big step forward to better understand the three-dimensional evolution of melt ponds during the melt season. Such data is also extremely valuable for validation and improvement of melt pond parameterisations in sea ice models. One example, the relation between pond depth and melt pond fraction, is given in the present paper.

We are glad to hear the reviewer found both the results and discussion of the limitations of the tested methods to be valuable, and that they agree that the resulting data set will be valuable for validation and improvement of sea ice melt pond parameterizations.

As cloud-free conditions are a prerequisite for retrieving melt pond properties from the three satellites used, it is still not possible to analyse melt pond evolution in a very high resolution in time in a wider area. However, airborne campaigns could bridge certain gaps. As outlined in the conclusions, such airborne observations as well as in situ observations (coincident with ICESat-2 passes) will also be very helpful to quantify the uncertainty from ICESat-2 measurements.
To summarize, the paper addresses very relevant scientific questions within the scope of TC, and it presents novel data. Substantial conclusions are reached. The scientific methods are clearly outlined and the results are sufficient to support the conclusions.
The title reflects the content of the paper, the abstract provides a complete summary and the paper is generally well structured. The review of existing published work is very good, the number of references is appropriate. Overall, figures and tables are clear and their captions self-explanatory. Mathematical formulae, symbols and abbreviations are correctly defined and used. The use of the English language is very good.

**Specific comments:**
It might be worth to add another aspect to the discussion. Based on the described errors and limitations in the melt pond properties retrieved from Sentinel-2 and ICESat-2, what is the impact of those errors and limitations on estimates of surface albedo?

We added a sentence to the Section 7 to acknowledge the impact of melt pond fraction results that are biased low on albedo estimates. Following the sentence: "Comparisons with higher-resolution WorldView images suggested that MPF estimates derived from Sentinel-2 are biased low by 7.2% on average, and up to 20% at the peak of the melt season (Sections 5.3 and 5.1)," we add: "Using these data for the derivation of albedo may lead to an overestimation of sea ice surface albedo, as an unponded surface has a higher albedo than a ponded surface."

**Technical corrections:**
Line 93/94: Please, check the years in "September 2020 average sea ice extent … (Fetterer et al., 2017)"
This is a dataset with the reference dated to 2017, but updated regularly. added "updated 2023"
Line 395: Please., check "decreased in".
change "in" to "from"
Line 407: "The the distributions …", delete "the".
deleted

---

## Author Comment (AC2)

Review of Buckley et al. Observing the Evolution of Summer Melt on Multiyear Sea Ice with ICESat-2 and Sentinel-2

We thank the reviewer for their comments and are happy to hear that they feel our assessment of the limitations of the Sentinel-2 imagery is both useful and insightful. Your suggestions and comments have improved the manuscript. The following responses (in blue font) address your comments point by point.

The authors provided a mainly methodological paper that integrates various remote sensing sources to track the evolution of melt pond characteristics (melt pond fraction and depth) on multiyear sea ice in the perennial ice zone north of the Canadian Arctic Archipelago in the 2020 melt season. This region contains some of the oldest and thickest ice in the Arctic and is distinct from the broader and largely first-year ice dominated Arctic pack. Multispectral optical data from Sentinel-2 and Worldview-2/3 satellites are used to estimate melt pond fraction (MPF) by image classification, and the resolution limitations of Sentinel-2 (10m) are evaluated in the context of the classifier by using the high-resolution Worldview-2/3 images (~1m) as verification. This is useful and insightful, as it provides new information about the limitations of the widely available but resolution constrained Sentinel-2 to estimate and study MPF broadly.

We thank the reviewer for their comment that our results provide details that will be essential for studies of atmosphere-ice-ocean processes.

As perhaps expected, the Sentinel-2 classifier underestimates MPF due to its inability to resolve smaller scale features such as small ponds and interconnecting meltwater channels due to pixel mixing. Despite being expected, it is good to see this problem being addressed rigorously and the limitations quantified. The authors found Sentinel-2 MPF is biased low "by up to 20.7% and averaging 7.2%, when small ponds are widespread across the surface". The authors also provide a method for adjusting the Sentinel-2 MPF based on input Worldview imagery when available. This approach is limited by the availability of high-resolution optical imagery, which should be less of a problem in the future as more open access data of that type becomes available.

Agreed, should more open access high-resolution optical imagery become available in the future, it should be used to quantify the biases in the lower-resolution, but more widespread, Sentinel-2 observations.

There is a sentence in the conclusions that addresses this: "The bias can be quantified and corrected using higher resolution WorldView imagery when available (Section 5.3)."

The authors further compare two novel melt pond depth retrieval algorithms that use ICESat-2 altimetry data as input (ATL03 geolocated photon height) and discriminate pond surfaces from bottoms to resolve pond depth. They are unique, such that a comparison like this provides detail essential to their effective application to studies of atmosphere-ice-ocean processes during advanced stages of sea ice melt, when melt ponds play a key role in energy and meltwater budgets, among other things. The limitations of the algorithms are explicitly and thoroughly addressed in the context of the detection limits of ICESat-2, e.g. the pulse width imposes a 0.2m

limit on pond depth, which means depth would not likely be resolved on seasonal sea ice due to their shallow nature. As the authors clearly indicate, one of the algorithms (DDA) is superior to the other (UMD-MPA) for detecting pond depth in a larger proportion of melt ponds in the studied area because it can resolve smaller ponds.

Sea ice and melt pond fractions are combined with melt pond depth estimates to examine the evolution of the advanced melt season in 2020 and the evolving parameters are discussed in the context of known key periods (early melt, maximum melt, etc.), previous observations from field campaigns SHEBA and MOSAiC, and model parameterizations (CESM and CICE). In general, the authors draw similarity between their results and those of others except when comparison to the modelling approaches where there is no observed relationship between pond depth and fraction.

The paper is original and presents new insights on the combined evolution of melt pond fraction and depth, thus providing an important third dimension to the study of these important features using remote sensing data. The methods are valid and suitable and connections to parallel and ongoing work on the melt pond depth algorithm (DDA) are clearly and helpfully stated. Enough detail is provided to understand both application potential and limitations of the described methods. The topic is definitely appropriate for The Cryosphere, though the authors should first address the below major and minor comments.

**Major comments**

The paper is imbalanced in that it presents as an analysis of the 2020 melt season due to its relevance in a climate context and in terms of the sea ice record – anomalously warm spring, early melt, and the second lowest September sea ice extent on record. It is more focused on the methods associated with melt pond fraction and depth retrieval and limitations. By comparison, the melt season analysis is rather cursory and mostly limited to a time series analysis of the derived parameters. The authors could strengthen the paper by either **placing more emphasis on the methodological components as the core theme of the paper (e.g. section 5.2.4 belongs in results),** or by expanding the melt season analysis so that the 2020 melt season evolution, as it is characterized by the retrieved melt pond properties, is better understood in the context of being an anomalous season. The latter would benefit from a comparison of 2020 conditions to other years which understandably would be limited to 2021 and 2022 by Sentinel-2 and ICESat-2 availability.

Thank you for the comment. This paper makes use of methodologies described in previous work: Buckley et al., 2020, and Herzfeld et al., 2019, 2023, and also expands on an initial feasibility study in Farrell et al., 2020. The study is the first time that algorithms designed to retrieve melt pond depth and melt pond fraction from satellite platforms have been applied to coincident altimeter data and optical imagery, and the derived results combined to present a fuller picture of the melt evolution. We consider this paper a study of the summer 2020 evolution, focusing on understanding the seasonal evolution of melt on multiyear ice and for the first time, simultaneously tracks changes in melt pond fraction and depth. Future work will analyze the 2021 and 2022 melt seasons that can enhance our understanding of melt progression and the interannual variability in melt evolution. We have added a sentence to the end of the conclusions: "Expanding this study to other melt seasons can provide information on the interannual variability of the melt evolution."

There is no section 5.2.4, we believe you are referring to 4.2.4 Algorithm Limitations. To address your comment, we have reorganized the paper to focus on the methodological components in Sections 3.3 and 4.2 and we have moved the discussion of the algorithm limitations into the results instead (now Section 5.4). This section includes the image pixel misclassifications (Section 5.4.1, previously section 3.4), the impact of lower resolution imagery on derived parameters (Section 5.4.2, previously section 3.5), and the melt pond depth algorithm limitations (Section 5.4.3, previously section 4.2.4). The melt pond size distribution analysis was moved to the results as well (now the pond size distribution results are presented in Section 5.3, previously section 4.2.5).

The figures have been reordered and discussion adjusted as necessary to accommodate these changes.

The authors should better describe the combined and relative importance of melt pond fraction and depth for the study of sea ice melt season evolution and atmosphere-ice-ocean interactions. This could be more explicitly addressed through the stated research goal(s) of the paper. Outlining the potential benefits of merging depth estimates with pond fraction, i.e. for volume estimates, and for improved understanding of melt evolution, is needed. How tracking the depth in only the larger ponds would influence the results is also of relevance. This is needed partly due to the imbalanced nature of the paper, as mentioned above. Detail regarding the importance of these observations mainly comes at the end of the conclusions, whereas the emphasis is on the methods early on and only general statements are made (e.g. line 83, understanding the evolution of melt).

Per the previous suggestion, we have rearranged the paper so that this is better organized. In Section 6.2 we discuss the relationship between pond fraction and depth, relying on the combined melt pond depth and fraction datasets. This study does not include calculations of meltwater volume and thus we mention the potential for melt pond volume estimates in the conclusion as an avenue for future research. Pond volume estimates are included in Farrell et al., 2020, demonstrating feasibility of combining the depth and area measurements.

We have added a sentence in image classification after defining MPF and SIC:
L167: "Understanding how MPF and SIC change throughout the summer melt season can provide insights about the evolution of surface albedo and the absorption of solar radiation."

And throughout the paper, we have emphasized the importance of melt pond fraction, depth, and volume estimates:

Instances where we discuss the importance of depth:
L195: "We are able to estimate pond depth, an important characteristic of melt ponds since it constrains meltwater volume and alters the hydrostatic balance of the sea ice."

On the melt pond area and albedo implications:
L15: "During the summer, highly reflective snow covered Arctic sea ice with an albedo > 0.7 decreases due to both the disintegration of the ice cover exposing the low-albedo open ocean

(albedo < 0.1) and melt ponding on the ice surface (albedo 0.1 to 0.3). This rapid change in albedo drives the positive ice albedo feedback enabling additional uptake of shortwave radiation, enhancing melt. "

On meltwater volume and fluxes into the ocean:
L18:"Meltwater percolation through the ice freshens the underlying ocean.."

Minor Comments (L=line)
L90: Clarify what area is the multiyear ice region.
clarified: "(Figure 1, purple)"
L120: Give the spectral range as done for Worldview below.
added: "ranging from 443 nm to 2190 nm"
L146: "Worldview"
edited
L168: Delete Maxar
done
L173: Pixel misclassification is discussed in section 3.3 and 3.4 and both discussions refer to the same figure. Should be in one section.
Thank you for this comment. These sections are now combined and in the results section 5.4 Algorithm Limitations. Section 5.4.1 discusses pixel misclassification, section 5.4.2 discusses the impact on derived parameters.
L178: Add "image" after Worldview.
added
L213: Use more precise terminology than chunks.
changed to "brash ice"
L218: Delete second comma
deleted
L229: Use MPF instead of melt pond fraction.
adjusted
L233: area "of" …
added 'of'
L238: Use SIC after Sentinel-2.
changed
L253: MPF
changed "melt pond fraction" to "MPF"
L259: Use ATLAS only, it was defined earlier.
adjusted sentence. "ATLAS has 3 beam pairs with 90-m spacing within the pairs…"
L371: "false"
corrected
L379: Does the dead time effect also happen for leads as it does for melt ponds?
It does, yes. Added to a sentence L503: "specular leads and melt ponds" and added reference (Kwok et al., 2019; Tilling et al., 2020)
L402: The Perovich et al. 2002b paper is cited, but a little more context about the study would still be appropriate.

Good suggestion. Since Perovich et al. (2002b) and Perovich et al. (2003) are heavily referenced throughout the manuscript, we adjusted the sentence in the introduction as follows:
"This study is motivated by the initial work observing melt pond evolution at the SHEBA site from aerial imagery acquired weekly (Perovich et al., 2002b) and regular melt pond depth measurements (Perovich et al., 2003) in 1998 in the Beaufort Sea. Here, we extend our understanding of the evolution of sea ice melt."

L406: How do we know that this is due to seasonal evolution versus differences in the area covered by the Worldview scenes and the ice topography etc.?

We do not know that this is just due to seasonal variation. We have added the following sentence: "However, we note that due to ice drift the images analyzed do not depict the same ice throughout the season, and although the melt ponds loosely follow the expected evolution of melt pond circularity, other factors such as ice topography and local ice and atmospheric conditions affect the evolution of melt ponds and their geometric features"

L429: "imagery"

changed to "imagery"

L436: Provide detail on how the presence of ice lids is determined. Is it the dark grey color mentioned, in which case can this be confused with a drained melt pond?

In this particular image (Figure 8e), there are ponds with only part of the surface frozen, which makes it clear it is not a drained pond. The smallest ponds are entirely grey, with frozen lid across the entire surface, whereas the larger ponds still have portions of the surface that remain unfrozen. Small ponds freeze up faster than large ponds due to their relatively lower heat capacity. See zoom in on the image here:

[Figure]

Also, the grayish color of a pond lid is similar to the nilas seen in the image on 3 September (Figure 8f). We clarified the text:

In some ponds, the surface or a portion of the surface of the pond had refrozen to form an ice lid, indicated by a dark gray color, similar to the color of nilas appearing in Figure 8f.

L555: Can delete ", and inability to track smaller ponds"

deleted

L614: "…being biased high…"

added "being"

Figure 6 caption: word missing "Examples (c) and (d) the subsurface". Show?

added "show"